# Clinical and Biomarker Characteristics According to Clinical Spectrum of Alzheimer’s Disease (AD) in the Validation Cohort of Korean Brain Aging Study for the Early Diagnosis and Prediction of AD

**DOI:** 10.3390/jcm8030341

**Published:** 2019-03-11

**Authors:** Jihye Hwang, Jee Hyang Jeong, Soo Jin Yoon, Kyung Won Park, Eun-Joo Kim, Bora Yoon, Jae-Won Jang, Hee Jin Kim, Jin Yong Hong, Jong-Min Lee, Hyuntae Park, Ju-Hee Kang, Yong-Ho Choi, Gilsoon Park, Jinwoo Hong, Min Soo Byun, Dahyun Yi, Yu Kyeong Kim, Dong Young Lee, Seong Hye Choi

**Affiliations:** 1Department of Neurology, Keimyung University Dongsan Medical Center, Daegu 41931, Korea; jh.hwang0110@gmail.com; 2Department of Neurology, Ewha Womans University School of Medicine, Seoul 07985, Korea; jjeong@ewha.ac.kr; 3Department of Neurology, Eulji University School of Medicine, Daejeon 35233, Korea; trumind@eulji.ac.kr; 4Department of Neurology, Dong-A Medical Center, Dong-A University College of Medicine, Busan 49201, Korea; neuropark@dau.ac.kr; 5Department of Neurology, Pusan National University Hospital, Pusan National University School of Medicine and Medical Research Institute, Busan 49241, Korea; eunjookim@pusan.ac.kr; 6Department of Neurology, Konyang University College of Medicine, Daejeon 35365, Korea; boradori3@naver.com; 7Department of Neurology, Kangwon National University Hospital, Kangwon National University College of Medicine, Chuncheon 24289, Korea; light26@hanmail.net; 8Department of Neurology, Samsung Medical Center, Sungkyunkwan University School of Medicine, Seoul 06351, Korea; evekhj.kim@samsung.com; 9Department of Neurology, Yonsei University Wonju College of Medicine, Wonju 26426, Korea; jinyhong@yonsei.ac.kr; 10Department of Biomedical Engineering, Hanyang University, Seoul 04763, Korea; ljm@hanyang.ac.kr (J.-M.L.); chldydgh0128@hanyang.ac.kr (Y.-H.C.); pks1207@hanyang.ac.kr (G.P.); jwhong1125@gmail.com (J.H.); 11Department of Health Care and Science, Dong-A University, Busan 49315, Korea; htpark@dau.ac.kr; 12Department of Pharmacology, Inha University School of Medicine, Incheon 22212, Korea; johykang@inha.ac.kr; 13Institute of Human Behavioral Medicine, Medical Research Center Seoul National University, Seoul 03080, Korea; bminsoo@gmail.com (M.S.B.); dahyunyi@gmail.com (D.Y.); 14Department of Nuclear Medicine, SMG-SNU Boramae Medical Center, Seoul 07061, Korea; yk3181@snu.ac.kr; 15Department of Neuropsychiatry, Seoul National University Hospital & Department of Seoul National University College of Medicine, Seoul 03080, Korea; selfpsy@snu.ac.kr; 16Department of Neurology, Inha University School of Medicine, Incheon 22332, Korea

**Keywords:** Alzheimer’s disease, mild cognitive impairment, biomarkers, cohort study, subjective cognitive decline

## Abstract

We aimed to present the study design of an independent validation cohort from the Korean Brain Aging Study for the Early Diagnosis and Prediction of Alzheimer’s disease (AD) (KBASE-V) and to investigate the baseline characteristics of the participants according to the AD clinical spectrum. We recruited 71 cognitively normal (CN) participants, 96 with subjective cognitive decline (SCD), 72 with mild cognitive impairment (MCI), and 56 with AD dementia (ADD). The participants are followed for three years. The Consortium to Establish a Registry for AD scores was significantly different between all of the groups. The logical memory delayed recall scores were significantly different between all groups, except between the MCI and ADD groups. The Mini-Mental State Examination score, hippocampal volume, and cerebrospinal fluid (CSF) amyloid-β42 level were significant difference among the SCD, MCI, and ADD groups. The frequencies of participants with amyloid pathology according to PET or CSF studies were 8.9%, 25.6%, 48.3%, and 90.0% in the CN, SCD, MCI, and ADD groups, respectively. According to ATN classification, A+/T+/N+ or A+/T+/N− was observed in 0%, 15.5%, 31.0%, and 78.3% in the CN, SCD, MCI, and ADD groups, respectively. The KBASE-V showed a clear difference according to the AD clinical spectrum in neuropsychological tests and AD biomarkers.

## 1. Introduction

Alzheimer’s disease (AD) is the most common cause of dementia and it is characterized by an accumulation of amyloid β (Aβ) and neurofibrillary tangles, which are associated with synaptic dysfunction and neurodegeneration that lead to memory impairment and other types of cognitive decline [1]. Recent studies suggest a 20- to 30-year interval between the first development of amyloid positivity and the onset of dementia [2,3]. The clinical spectrum of AD includes subjective cognitive decline (SCD), mild cognitive impairment (MCI), and AD dementia (ADD) [3]. SCD in individuals with unimpaired performance on cognitive tests is considered to represent the first symptomatic manifestation of AD [4]. Meta-analysis suggests that older people with SCD are twice as likely to develop dementia as individuals without SCD, and each year approximately 2.3% and 6.6% of elderly individuals with SCD may progress to dementia and MCI [5]. Due to AD, amnestic MCI with gradual and progressive episodic memory impairment is likely to be prodromal AD or MCI [6,7].

Despite extensive research in the AD field, no treatment has yet been developed to modify AD progression. Therefore, it is important to manage modifiable risk factors in the prevention of dementia [8]. Further research is also needed to find new risk, protective, and prognostic factors for AD.

Over the past decade, remarkable advances in biomarker identification have been made, including neuroimaging, cerebrospinal fluid (CSF) analysis, and others. Now, these biomarkers provide the ability to detect evidence of the AD pathophysiological process in vivo. The biomarkers associated with Aβ plaques include cortical amyloid positron emission tomography (PET) ligand binding [9,10,11] or low CSF Aβ42 [12]. The biomarkers that are indicative of fibrillar tau are elevated CSF tau phosphorylated at Thr181 (p-tau) and cortical tau PET ligand binding [13,14]. Biomarkers of neurodegeneration or neuronal injury are elevated CSF total tau (t-tau) [14], temporo-parietal or precuneus hypometabolism on fluorodeoxyglucose PET [15], and cortical or hippocampal atrophy on magnetic resonance imaging (MRI) [16]. Recently, the National Institute on Aging and Alzheimer’s Association (NIA-AA) Research Framework suggested ATN classification for the diagnosis of AD using biomarkers for studies with specific goals [14].

PET, MRI, and CSF biomarkers are very sensitive; however, the PET and MRI methods are costly and CSF studies can be complicated by headaches and back pain. Therefore, other cost-effective and noninvasive methods for the early diagnosis and prediction of AD are needed. The Korean Brain Aging Study for the Early Diagnosis and Prediction of AD (KBASE) is a prospective cohort study that began in Seoul and surrounding areas in 2014 and it is aimed at recruiting cognitively unimpaired (CU) individuals and patients with MCI or ADD to identify new biomarkers, particularly blood biomarkers, in the early detection of AD and risk factors for AD [17]. The validation cohort of KBASE (KBASE-V) is an independent nationwide cohort to reconfirm new potential biomarkers that were identified in the KBASE study and is intended to find risk and prognostic factors for AD as additional studies. The KBASE-V also consisted of participants with ADD, MCI, or CU, and the CU individuals were divided into SCD and cognitively normal (CN) individuals without SCD. We aim to present the study design of KBASE-V and investigate the baseline characteristics of participants according to the AD clinical spectrum.

## 2. Materials and Methods

### 2.1. Participants

We have recruited 71 CN participants, 96 SCD, 72 MCI, and 56 ADD in nine memory clinics across South Korea from April 2015 to August 2016. All of the participants were aged from 55 to 90 years with a reliable informant who could provide investigators with the requested information. All of the CN and SCD participants had normal age-, gender-, and education-adjusted performance on the memory tests of the Korean version of Consortium to Establish a Registry for AD (CERAD) (word list immediate recall, word list delayed recall, word list recognition, and constructional praxis recall) [18] and global Clinical Dementia Rating (CDR) scale scores of 0 [19]. The SCD participants met the SCD criteria that were established by the SCD-Initiative group [4]. The participants with MCI met the core clinical criteria for MCI due to AD established by the NIA-AA workgroups [7], and the following criteria that were modified from those proposed by Petersen et al. [20]: (1) CDR of 0.5; (2) memory complaints by patients, caregivers, or clinician in comparison with the participant’s previous level; (3) a performance score that was worse than 1.5 standard deviations (SD) below age, education, and gender-adjusted normative means for at least one of the memory tests included in the CERAD (word list immediate recall, word list delayed recall, word list recognition, and constructional praxis recall) [18]; (4) being capable of performing independent activities of daily living (ADL); and, (5) not demented. The ADD participants met the following inclusion criteria: (1) criteria for dementia according to the Diagnostic and Statistical Manual of Mental Disorders 4th Edition (DSM-IV-TR) [21]; (2) criteria for probable ADD according to the NIA-AA core clinical criteria [22]; and, (3) CDR score of 0.5 or 1. Exclusion criteria for all of the participants included: (1) the presence of major psychiatric illness; (2) significant neurological or medical condition or comorbidities that could affect cognitive functioning; (3) contraindications for MRI scan (e.g., pacemaker, claustrophobia); (4) illiteracy; (5) significant visual or hearing difficulty or severe communication or behavioral problems that would make a clinical examination or brain scan difficult; (6) taking an investigational drug; and, (7) pregnant or breastfeeding. The inclusion and exclusion criteria for ADD, MCI, and CU in the KBASE-V were the same as those that were in the KBASE. In addition, the CU group was divided into the CN group and the SCD group while using the SCD criteria that were established by the SCD-Initiative group [4] in the KBASE-V.

The study was performed in accordance with the International Harmonization Conference guidelines on Good Clinical Practice and the institutional review board of each center approved it prior to beginning the study (INHAUH 2015-03-021). Prior to participation in the study, all of the participants with CN, SCD, and MCI provided written informed consent to participate in the study, and all ADD patients, as well as their legal representatives, provided written informed consent to participate in the study.

### 2.2. Clinical Assessment

All of the participants underwent physical and neurological examinations and thorough diagnostic procedures, including an interview regarding the participants’ cognition, abnormal behaviors, ADL, demographic characteristics, vascular risk factors, current medication, family history, and other comorbidities; the Mini-Mental State Examination (MMSE) in the Korean version of CERAD assessment packet [18]; the Cognitive Complaint Interview (CCI) [23], Memory Assessment Clinics Questionnaire (MAC-Q) [24], and the Subjective Memory Complaints Questionnaire (SMCQ) [25] in the assessment of subjective cognitive complaint; the Geriatric Depression Scale (GDS) [26]; the Blessed Dementia Scale-ADL (BDS-ADL) [27]; the CDR and Global Deterioration Scale [19]; and the neuropsychiatric inventory (NPI) [28].

We measured height, weight, handgrip, hip and waist circumferences, mobility and balance using gait speed, the Timed Up and Go Test [29], and the one leg standing test [30]. The Parkinsonian symptoms were evaluated using the Unified Parkinson’s Disease Rating Scale [31]. We also conducted several questionnaires and scales to investigate the modifiable lifestyle risk factors, and these assessments included the following: the International Physical Activity Questionnaire [32]; simple appetite questionnaire [33], Eating Behavior Scale [34], and Mini Nutritional Assessment (MNA) [35] for evaluating nutritional status; the Pittsburgh Sleep Quality Index [36], Stanford Sleepiness Scale [37], and Epworth Sleepiness Scale [38] for sleep evaluation; and the Cardiovascular Health Study Frailty Index [39]. Sarcopenia was diagnosed according to the Asian Working Group criteria [40]. The above clinical evaluations, such as those that were performed at baseline, were repeated annually, and the NPI and UPDRS were performed every two years.

All of the participants underwent bioelectrical impedance analysis to measure the appendicular skeletal muscle mass index (ASMI) [40] and the proportion of fat every year. Participants also underwent laboratory tests every two years, including a complete blood count; blood chemistry; lipid panel; erythrocyte sedimentation rate; serum levels of vitamin B12, folate, 25-hydroxy vitamin D, and brain-derived neurotrophic factor (BDNF); C-peptide; glycated hemoglobin; adiponectin; homocysteine; the venereal disease research laboratory test; thyroid function test; routine urine analysis; and urine microalbumin. Apolipoprotein E (APOE) genotyping and BDNF polymorphism (BDNF Val66Met) status were evaluated at the baseline. Actigraphy was only conducted at baseline for the participants who agreed to the assessment.

### 2.3. Neuropsychological Assessment

All of the participants underwent comprehensive neuropsychological testing, including the CERAD [18], Stroop test (Korean Golden version) [41], clock drawing test, and Short Blessed Test (SBT) [42] every year. The following more detailed neuropsychological tests were performed every two years: the Trail Making Test (TMT) A and B [43], digit span forward and backward tests [44], Wechsler Memory Scale-Fourth edition Korean version (WMS-IV-K) Logical Memory (LM) I, II, and recognition [45], Rey Complex Figure Test (RCFT) copy, 3 min and 30 min recall tests of RCFT [46], the Wechsler Adult Intelligence Scale-Fourth edition Korean version (WAIS-IV-K) block design [47], Controlled Oral Word Association Test (COWAT), and Frontal Assessment Battery (FAB) [48].

### 2.4. Brain MRI

Brain MRI data were acquired from all of the participants every two years using a 3.0 T MR scanner, which captured three-dimensional (3D) T1-weighted and T2-weighted SPACE sagittal images of 0.8-mm thickness without gap, as well as axial fluid-attenuated inversion recovery, diffusion tensor imaging, and resting-state functional MRI. The MRI protocols were based on the Alzheimer’s Disease Neuroimaging Initiative phase 2 (ADNI-2) MRI protocols [49]. The 3D T1-weighted MRI parameters were as follows: TR = 7.32 ms, TE = 3.02 ms, TI = 400 ms, Flip Angle (FA) = 11°, and the voxel resolution was 0.8 × 0.8 × 0.8 mm^3^ in the General Electric (GE) Discovery MR750 scanner (GE Healthcare, Milwaukee, WI, USA); TR = shortest (6.8 ms), TE = shortest (3.1 ms), FA = 9°, and voxel resolution was 0.8 × 0.8 × 0.8 mm^3^ in the Achieva scanner (Philips Healthcare, Andover, MA, USA); and TR = 2300 ms, TE = 2.14 ms, TI = 900 ms, FA = 9° and voxel resolution was 0.8 × 0.8 × 0.8 mm^3^ in the Skyra and Trio Tim scanners (Siemens, Washington, DC, USA). The images were then forwarded to the laboratory at Hanyang University where the analysis was conducted.

The 3D T1-weighted MRI data were processed using CIVET pipeline version 2.1 (http://mcin-cnim.ca/neuroimagingtechnologies/civet/) [50]. The N3 intensity nonuniformity correction algorithm was used to calibrate the intensity difference that was caused by an inhomogeneity in a magnetic field [51]. Corrected T1-weighted images on native space were aligned to the Montreal Neurological Institute 152 standard space [52]. Nonbrain tissue was excluded while using the BET algorithm to focus on the brain [53]. The registered images were classified as gray matter, white matter, and CSF while using an advanced neural-net classifier [54]. The inner surfaces of the cortex were extracted from the partial volume corrected white matter mask using deformable spherical mesh, and then the outer surface of the cortex was automatically extracted using the constrained Laplacian-based automated segmentation with the proximities algorithm [55]. Cortical thickness values in native space were calculated using the Euclidean distance between the linked vertices of the inner and outer surfaces [56].

After intensity inhomogeneity correction, the corrected T1-weighted images were segmented into the left and right sides of the hippocampus using FMRIB’s Integrated Registration and Segmentation Tool (FIRST) [57]. The hippocampal volumes were normalized for total intracranial volume, defined as the sum of gray matter, white matter, and CSF volumes. The optimal cutoffs for the hippocampal volume and mean cortical thickness were generated with a ROC analysis to ascertain the optimal threshold for the sensitivity and specificity calculation using data from the ADD and CN groups. The cutoff values that yielded the best Youden’s index (sensitivity + specificity − 1) for ADD diagnosis were 3.07 mm (sensitivity 83.3%, specificity 85.9%) for mean cortical thickness and 4.67 cm^3^ (sensitivity 67.3%, specificity 80.3%) when averaging right and left hippocampal volumes.

### 2.5. Amyloid Positron Emission Tomography

Eighty participants underwent three-dimensional ^11^C-Pittsburgh Compound B (PiB) PET and 3D T1-weighted MRI using a 3.0 T Biograph mMR (PET-MR) scanner (Siemens, Washington, DC, USA) at baseline. After intravenous administration of 555 MBq of 11C-PiB (range, 450–610 MBq), 40 min after injection, a 30-min emission scan was obtained. The data were reconstructed into a 256 × 256 image matrix using iterative methods (six iterations with 21 subsets) and were corrected for uniformity, ultrashort echo time-based attenuation, and decay reduction. We aligned the ^11^C-PiB PET images to the corresponding T1-weighted MRI using rigid body transformation. The standard uptake value ratio (SUVR) of each region of interest (ROI) was obtained by dividing the mean value for all voxels within the ROI by the mean cerebellar gray matter uptake value. The Desikan-Killiany-Tourville (DKT) protocol atlas was used as the definition of ROIs for quantitative image analysis while using SUVR [58]. Composite SUVR values were formed by averaging of the SUVR values for the frontal, parietal, temporal, occipital, anterior cingulate, and posterior cingulate/precuneus cortices [9]. The cutoff value that yielded the best Youden’s index for ADD diagnosis from the CN group was 1.21 (sensitivity 84.6%, specificity 100%) for the composite cortical PiB retention SUVR. This cutoff value was the same as the previous study from the KBASE [59].

A total of 135 participants underwent ^18^F-flutemetamol PET, which included a 20 min PET scan (4 × 5 min dynamic frames), starting at 90 min after intravenous injection with 185 MBq ± 10% of ^18^F-flutemetamol, followed by a 10-mL saline flush. The data was acquired using Biography MCT PET/CT scanner (Siemens), Discovery 690 PET/CT scanner (GE), Discovery 600 PET/CT scanner (GE), Discovery STE PET/CT scanner (GE), or Gemini TF PET/CT scanner (Philips). Low-dose computed tomography was performed for attenuation correction before all of the scans. The images were reconstructed using an ordered subsets expectation maximization algorithm with four iterations and 16 subsets. Similar to PiB PET, the ^18^F-flutemetamol PET images were transformed to the corresponding MRI. SUVR was obtained by using the pons as a reference region and then parcellated using the DTK protocol atlas [58]. Composite SUVR values were calculated by a simple average of the SUVR values for the frontal, parietal, temporal, occipital, anterior cingulate, and posterior cingulate/precuneus cortices [11,60]. The cutoff value that yielded the best Youden’s index for distinguishing ADD from the CN participants was 0.634 (sensitivity 87.5%, specificity 93.3%) for the composite SUVR of cortical flutemetamol retention [11,60]. The ^11^C-PiB PET and ^18^F-flutemetamol PET images were transferred to Hanyang University’s laboratory and then analyzed.

### 2.6. Cerebrospinal Fluid Analysis

At baseline, the 139 participants who agreed to participate in the CSF study underwent lumbar puncture in the morning. CSF was collected in 15-mL polypropylene transfer tubes (Falcon, Corning Science, NY, USA) and then immediately centrifuged at 2000× *g* for 10 min at room temperature (RT). The supernatant (~10 mL) was frozen on dry ice and then transferred to a laboratory at Inha University where CSF biomarkers were measured. The shipped CSF samples were gently mixed with a pipette with a polypropylene tip after thawing at RT, and 0.4-mL CSF sample aliquots were frozen in polypropylene tubes (Sarstedt AG & Co., Nümbrecht, Germany) and then stored at −80 °C until analysis. Aβ42, t-tau, and p-tau were measured using the multiplex xMAP Luminex platform with INNO-BIA AlzBio3 kits (Fujirebio Europe, Ghent, Belgium). To minimize the effects of analytical variability on the results [61], CSF analysis was performed following the manufacturer’s instructions and unified standard procedures, as previously described [62,63]. We determined the analytical precision using nine pooled CSFs and found that the mean (SD) between-run precision (percent coefficient of variation (%CV) through five runs) for CSF Aβ42, t-tau and p-tau was 9.1 (2.0), 10.9 (2.5), and 10.7 (6.2) %. All of the runs followed our standard operating procedure, including the pre-defined run acceptance criteria, including enough bead count (>20) and low %CV (<25%) in duplicates of calibrators, aqueous QC, and pooled CSFs.

The cutoff values that provided the highest Youden’s index for ADD diagnosis from the CN group were 433.68 pg/mL (sensitivity 91.3%, specificity 89.7%) for Aβ42, 57.77 pg/mL (sensitivity 69.6%, specificity 82.8%) for t-tau, and 22.55 pg/mL (sensitivity 78.3%, specificity 93.1%) for p-tau.

### 2.7. ATN Classification

According to the ATN classification scheme [14], each participant was classified into three binary categories. A+ refers to Aβ pathology (CSF Aβ42 levels ≤ 433.68 pg/mL or SUVR on ^18^F-flutemetamol PET ≥ 0.634 or SUVR on ^11^C-PiB PET > 1.21), T+ refers to pathologic p-tau (CSF p-tau ≥ 22.55 pg/mL), and N+ refers to the neurodegeneration biomarker (hippocampal volume ≤ 4.67 cm^3^ or mean cortical thickness ≤ 3.07 mm).

### 2.8. Statistical Analysis

For comparison among the groups, we used chi-square tests for categorical variables and one-way analysis of variance (ANOVA) for continuous variables. When a statistical significant overall difference was detected in the ANOVA test, pairwise comparisons on means between the diagnosis groups were conducted by the Tukey method. Analysis of covariance (ANCOVA) was performed for statistical analysis of Z scores of neuropsychological tests, with GDS score as a covariate. The Z scores were based on the mean and SD of each measure in the age-, sex-, and education-matched normal Korean elderly population that was reported in previous studies [18,41]. The CERAD Z score was calculated by averaging the Z scores of the CERAD subtests. GDS score was compared using ANCOVA, with age, sex, and education level as covariates. ANCOVA was also performed for statistical analysis of BMI, ASMI, hand grip, gait speed, MNA, total sleep time, ESS, CCI, MMSE, SBT, SMCQ, MAC-Q, NPI, BDS-ADL, CDR-SB, and raw scores of neuropsychological tests with age, sex, education level, and GDS as covariates. When a statistical significant overall difference was detected in the ANCOVA test, pairwise comparisons on the means between diagnosis groups were conducted by Bonferroni post hoc analysis for the correction of multiple comparisons. To compare the continuous clinical variables between each group of KBASE-V and its corresponding group of KBASE, ANCOVA tests were conducted with sex and education level as covariates in the CU group, sex as a covariate in the MCI group, and education level as a covariate in the ADD group. The statistical analyses were performed while using SPSS 19.0 (SPSS, Chicago, IL, USA). Values of *p* < 0.05 were considered to be statistically significant.

## 3. Results

The same measures with the KBASE and the additional measures of the KBASE-V among those that were performed in the KBASE-V are shown in Table 1. The neuropsychological tests and clinical evaluations that were performed in the KBASE-V were basically the same as the KBASE and additional evaluations of lifestyle risk factors were made in the KBASE-V.

There was no significant difference in age, MMSE, SBT, CDR, Stroop test, clock drawing, and the prevalence of the APOE ε4 carriers and A+ between each clinical group and its corresponding group in the KBASE (Table 2). The education level in the CU and ADD groups of KBASE-V was lower than the corresponding group of KBASE. In comparison with the corresponding group of KBASE, the prevalence of women was higher in the KBASE-V CU group and lower in the KBASE-V MCI group. The MCI group of KBASE-V had higher CERAD and lower CDR-SB scores than the MCI group of KBASE. The CU and ADD groups of KBASE-V showed higher GDS and SMCQ scores than the corresponding group in the KBASE.

The demographics and baseline characteristics of the KBASE-V participants, according to the AD clinical spectrum, are presented in Table 3. Age was significantly higher in the MCI and ADD groups than in the CN and SCD groups. The prevalence of women was higher in the CU and ADD groups than in the SCD and MCI groups. Education level, body mass index (BMI), MNA scores, and the prevalence of current drinkers and participants that meet the WHO physical activity (PA) guidelines (≥600 metabolic equivalent minutes of PA per week) [64] were lower in the ADD group than in the other groups. The MMSE, SBT, and CDR-SB scores were significantly different between all of the groups, except between the CN and SCD groups (Figure 1A). The CERAD scores differed by diagnostic group with better performance in the direction of CN > SCD > MCI > ADD (Figure 1B). The CCI, MAC-Q, SMCQ, and GDS scores of the SCD group were not different from those of the MCI group, but they were significantly higher than those in the CN group. The NPI and BDS-ADL scores and the prevalence of APOE ε4 carriers and sarcopenia were significantly higher in the ADD group than in the other groups.

The scores of naming, immediate recall of LM, constructional praxis recall, and 30 min delayed recall of RCFT were significantly different among the four groups with better performance in the direction of CN > SCD > MCI > ADD (Table 4). The scores of verbal fluency, digit span backward, correct color reading of Stroop test, and TMT-B in the SCD group were no different from those in the MCI group, but they were significantly lower than in the CN group. The scores of word list immediate and delayed recalls, word list recognition, 3 min delayed recall of RCFT, and FAB were significantly different between all of the groups, except between the CN and SCD groups. The LM delayed recall scores were significantly different between all of the groups, except between the MCI and ADD groups (Figure 1C). The RCFT copy and block design scores were significantly lower in the ADD group than in the other groups. The scores of constructional praxis, clock drawing, COWAT, and LM recognition in the CN group were significantly higher in those of the ADD and MCI groups, but they were not different from those in the SCD group.

84.2% of participants with ADD presented evidence of brain amyloid pathology on amyloid PET, 41.1% of those with MCI, 23.1% of those with SCD, and 4.5% of the CN individuals (Table 5). The prevalence rates of participants with A+ in the CN, SCD, MCI, and ADD groups were 8.9%, 25.6%, 48.3%, and 90.0%, respectively. The composite SUVRs on ^11^C-PiB PET and on ^18^F-flutemetamol PET were significantly higher in the ADD group than in the other groups. The composite SUVRs on ^18^F-flutemetamol PET in the MCI group was also higher than those of the CN group (Figure 1D). The CSF Aβ42 levels and hippocampal volume were significantly different between all of the groups, except between the CN and SCD groups (Figure 1E,F). CSF Aβ42 levels and hippocampal volume were lowest in the participants with ADD, the highest in the CN participants, and intermediate in those with MCI. CSF t-tau, p-tau, t-tau/ Aβ42, and p-tau/ Aβ42 levels were significantly higher in the ADD group than in the other groups (Table 5). The mean cortical thickness was significantly smaller in the ADD group than in the MCI and CN groups.

The 139 participants with CSF AD biomarker results were classified according to the ATN criteria (Table 6 and Figure 2). The proportion of participants in the Alzheimer’s continuum according to biomarkers was 10.3% in the CN group and it included one (3.4%) participant with preclinical Alzheimer’s pathologic change (A+T−N−) and two (6.9%) with Alzheimer’s and concomitant suspected non Alzheimer’s pathologic change (SNAP) (A+T−N+). The proportion to participants in the Alzheimer’s continuum was 24.1% in the SCD group and it included one (1.7%) participant with preclinical Alzheimer’s pathologic change, nine (15.5%) with preclinical AD (A+T+N−, A+T+N+), and four (6.9%) with Alzheimer’s pathologic change and SNAP (A+T−N+). The proportion of participants in the Alzheimer’s continuum was 55.1% in the MCI group and it included three (10.3%) participants with Alzheimer’s pathologic change with MCI, nine (31.0%) with prodromal AD (A+T+N−, A+T+N+), and four (13.8%) with Alzheimer’s pathologic change and SNAP with MCI (A+T−N+). The proportion of participants in the Alzheimer’s continuum was 91.3% in the ADD group and it included 18 (78.3%) participants with ADD (A+T+N−, A+T+N+) and three (13.0%) with Alzheimer’s pathologic change and SNAP with dementia (A+T−N+).

## 4. Discussion

Each clinical group of KBASE-V displayed generally comparable baseline characteristics with its corresponding group of KBASE, with some differences. General cognitive function, such as MMSE, SBT, and CDR in each clinical group was similar between the KBASE and KBAE-V. The age and the prevalence of APOE ε4 carriers were also similar between each clinical group of KBASE-V and its corresponding group of KBASE. The education level in the CU and ADD groups of KBASE-V was lower than the corresponding group of KBASE. The cause is that the KBASE-V is a nationwide cohort and that the KBASE is a cohort of highly educated people that live in the capital and surrounding areas. A higher GDS score in the CU and ADD groups of KBASE-V may also be related to the socioeconomic differences between KBASE and KBAE-V. The CERAD score in the CU and ADD groups of KBASE-V was similar to that in the corresponding group of KBASE. The cognitive function of the MCI group of KBASE-V appears to be slightly better than the corresponding group of KBASE, as the MCI participants in KBASE-V displayed higher CREAD and lower CDR-SB scores than the MCI group in KBASE. The prevalence of A+ was similar between each clinical group of KBASE-V and its corresponding group of KBASE. A total of 216 (73.2%) participants underwent amyloid PET in the KBASE-V. The presence or absence of amyloid pathology according to CSF analysis or amyloid PET was confirmed in 229 (77.6%) participants of the KBASE-V. The percentage of participants in KBASE-V for who the amyloid biomarkers were examined was not less than in the previous cohort [49,65]. The KBASE-V is a suitable cohort for reconfirming potential new AD biomarkers, such as blood biomarkers that were identified in the KBASE and identifying the risk and prognostic factors of AD.

The amyloid positivity rates in ADD were similar to those that were observed in the ADNI [65]. Amyloid positivity rates in MCI were similar to those in the previous study [66], but they were less than in the ADNI [65]. There may have been several factors contributing to this difference. First, the percentage of subjects who were APOE ε4 carriers in the present study (22% of MCI participants) is lower than in the APOE ε4-enriched multi-center research studies, such as ADNI (54% of MCI participants) [65] and the Australian Imaging, Biomarker, and Lifestyle study (55% of MCI participants) [67]. Assuming that APOE ε4 is closely related to increased amyloid positivity, the lower rate of ε4 carriers in this study is consistent with the lower rate of amyloid positivity in this study. Second, the inclusion criteria of MCI may have contributed to the lower observed rate of amyloid positive cases. We included the participants with poor performance in at least one of word list immediate recall, word list delayed recall, word list recognition, and visual memory test, in the MCI group. Therefore, it is likely that the MCI participants in the present study represented more heterogeneous group of individuals who, on average, had milder levels of impairment. Such individuals might be more typical of those who would present in a clinical setting for initial diagnosis.

Amyloid positivity rates (19.8%) in the CU participants, including the CN and SCD participants, were lower than in the ADNI, but were similar to those that were reported in the Japanese ADNI [65]. The difference may be due to age differences between the ADNI and KBASE-V. The CU participants of ADNI were older than those in the KBASE-V. The proportion of the CU participants was as high as 167 (56.6%) in the KBASE-V. Thus, the KBASE-V is well suited to investigate the natural history and prognostic factors of preclinical AD. The CSF AD biomarkers were investigated in 139 (47.1%) participants, and the clinical features and natural course according to ATN classification can be studied in CU and other groups.

In this study, the CN and SCD groups showed very different clinical features. The delayed recall tests for episodic memory and language function were significantly worse in the SCD group than in the CN group. However, visuospatial function and executive function test scores, except for the Stroop test, were not different between the SCD and CN groups. These findings are similar to those of a previous report [68]. General cognitive function, such as MMSE and CDR-SB, was not different between the CN and SCD groups. These results were also compatible with those from previous studies [68,69]. Depression scale scores in the SCD group were not different from those in the MCI and AD groups, but they were significantly higher than those in the CN group. The cause of SCD may be depression in some participants [70]. Alternatively, depression may be an early detection marker of AD [71]. We included the depression score as a covariate in the analysis to compare the clinical and neuropsychological variables among the groups, because depression was a key covariate in this group comparison. In the future, the comparison of clinical course between SCD with depression and SCD without depression is needed in a long-term follow-up study.

In this study, the subjective memory questionnaires could not differentiate between the SCD and MCI groups. The SCD individuals may complain of a memory impairment that is similar to that reported by MCI patients. The CDR-SB scores began to significantly increase from the MCI stage and ADL impairment began at the mild ADD stage. Therefore, CDR-SB and ADL are less likely to be used as a measure of drug efficacy in SCD and they may be useful in MCI [72]. Abnormal behaviors on the NPI and visuospatial dysfunction were noticeable from the mild dementia stage. The delayed recall score of LM did not differ between the MCI group and ADD group after correction for multiple comparisons. The LM delayed recall test appears to show a floor effect, even at the mild ADD stage, since the LM delayed recall score was almost zero in the participants with mild ADD of a CDR 0.5 or 1 in the KBASE-V. The results are similar to those of the previous report [73]. Therefore, the LM tests may not be suitable in monitoring the natural course of ADD.

BMI and nutritional state were decreased in the ADD group when compared with the other groups. Approximately 29% in this mild ADD group had sarcopenia, which is compatible with the results from a previous study [74]. Sarcopenia has been linked to higher levels of physical limitation and disability in older people [40]. As dementia progresses, significant impairment of nutritional indicators is observed [75]. We need to identify and treat sarcopenia and malnutrition to prevent disability in patients with dementia.

The cutoff of CSF Aβ42 levels for distinguishing ADD from the CN participants in the KBASE-V was much higher than in the ADNI [65]. The possible causes of the discrepancy with the ADNI were as follows. First, it is well recognized that interlaboratory variability in the immunoassay-based measurement of CSF Aβ42 is very large [76]; hence, the interlaboratory variability might be a reason of higher cut-off value in our study. Second, a small sample size may be related to the higher cutoff value of CSF Aβ42 levels. Third, it may be associated with age difference between the ADNI and KBASE-V. The participants in the KBASE-V were younger than those in the ADNI. Fourth, the mean score of the CSF Aβ42 levels in each group of the KBASE-V was higher than in the corresponding group of ADNI, but it was similar to that in the corresponding group of Japanese ADNI [65]. There may be ethnic difference in CSF Aβ42 levels. Finally, we could not exclude a technical error. The possibility seems to be low, since we determined the acceptable precision of xMAP-Luminex with AlzBio3 kit (average between-run %CV for Aβ42 through five runs was 9.1 (2.0) %) using pooled nine CSFs before the measurement of our samples.

Although not being statistically significant, except for the comparison between the ADD group and the other groups, CSF t-tau and p-tau and the accumulation of amyloid on amyloid PET increased as the clinical spectrum of AD progressed. CSF Aβ42 levels were significantly decreased from SCD to MCI and ADD. This is compatible with a previous report [77]. Cortical atrophy was more severe in the SCD group when compared with that in the CN group. Hippocampal atrophy seems to be prominent, beginning at the MCI stage. According to the ATN classification, the proportion of those in the Alzheimer’s continuum increased as the clinical spectrum of AD progressed.

There are some limitations in this study. First, the number of participants was relatively small. In particular, the number of MCI and ADD participants was small. Second, the amyloid PET or CSF biomarker studies were not conducted in all patients. Third, the CSF AD biomarker study was only conducted in approximately half of the participants. Therefore, ATN classification can be applied in approximately half. Fourth, the sensitivity of CSF t-tau was low. We have included cortical atrophy and hippocampal atrophy on brain MRI in the neurodegeneration markers.

In conclusion, we constructed a KBASE-V cohort consisting of CN, SCD, MCI, and ADD participants, which are followed-up for three years. This cohort showed clear differences in neuropsychological tests and AD biomarkers, according to the clinical spectrum of AD. This cohort is suitable to reconfirm the potential new AD biomarkers that were identified in the KABSE. It is also useful to find the risk and prognostic factors for AD and to evaluate the natural course of disease according to the AD clinical spectrum.

## Figures and Tables

**Figure 1 jcm-08-00341-f001:**
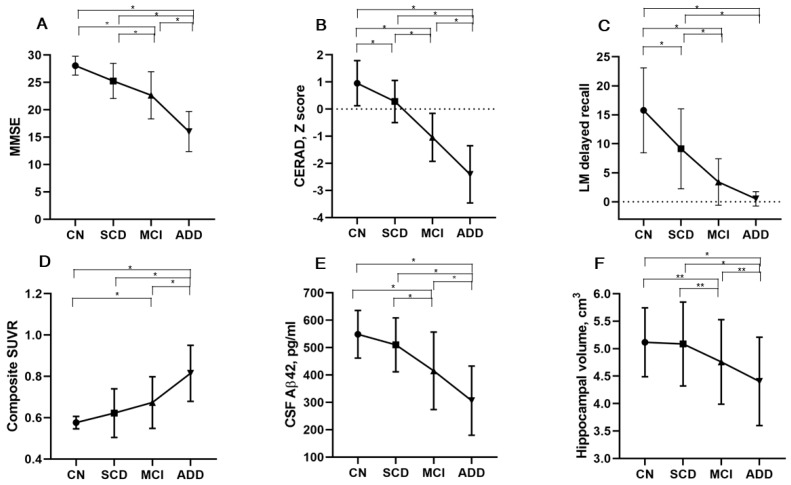
Mean (SD) of representative imaging and cognitive summary measures according to the Alzheimer’s disease (AD) clinical spectrum. In the Mini-Mental State Examination (MMSE) scores, CSF Aβ42 level and hippocampal volume, there was a significant difference between all the groups, except between cognitively normal (CN) and subjective cognitive decline (SCD) groups (**A**, **E**, and **F**). The Consortium to Establish a Registry for AD (CERAD) Z scores were significantly different between all groups with better performance in the direction of CN group > SCD > mild cognitive impairment (MCI) > AD dementia (ADD) (**B**). Mean score of the logical memory (LM) delayed recall test differed between all groups, except between MCI and ADD groups (**C**). The composite standardized uptake value ratio (SUVR) of cortical flutemetamol retention was significantly different between all groups, except between CN and SCD and between SCD and MCI (**D**). * *p* < 0.01; ** *p* < 0.05.

**Figure 2 jcm-08-00341-f002:**
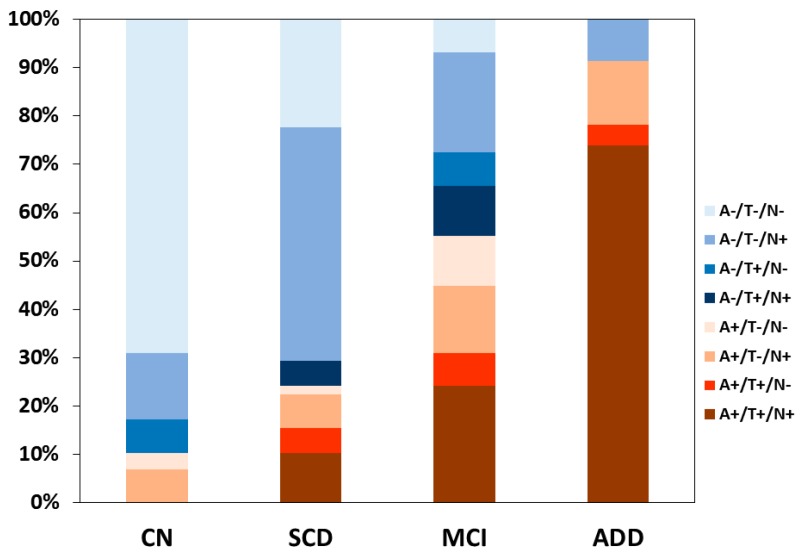
A stacked bar graph showing the proportions of ATN categories within each diagnostic group among the KBASE-V participants with cerebrospinal fluid (CSF) AD biomarker results. The proportion of participants in the Alzheimer’s continuum (A+/T+/N+, A+/T+/N−, A+/T−/N+, and A+/T−/N−) was 10.3% in the CN group, 24.1% in the SCD group, 55.1% in the MCI group, and 91.3% in the ADD group. CN, cognitively normal; SCD, subjective cognitive decline; MCI, mild cognitive impairment; ADD, Alzheimer’s disease dementia; ATN, amyloid, tau, and neurodegeneration or neuronal injury; A−, absence of amyloid pathology determined by normal amyloid PET finding or CSF Aβ42 level above cut point; A+, presence of amyloid pathology determined by abnormal amyloid PET finding or CSF Aβ42 level below cut point; T−, normal CSF p-tau level below cut point; T+, abnormal CSF p-tau level above cut point; N−, absence of neurodegeneration or neuronal injury determined by cortical thickness or hippocampal volume; N+, presence of neurodegeneration or neuronal injury determined by cortical thickness or hippocampal volume.

**Table 1 jcm-08-00341-t001:** The same measures with the Korean Brain Aging Study for the Early Diagnosis and Prediction of AD (KBASE) and the additional measures of the validation cohort of KBASE (KBASE-V).

Same Measures with the KBASE	Additional Measures in the KBASE-V
Inclusion and exclusion criteria Mini-Mental State Examination [18] Consortium to Establish a Registry for AD [18] Short Blessed Test [42] Detailed neuropsychological tests: digit span forward and backward tests [44]; logical Memory test [45]; Rey Complex Figure Test [46]; block design [47]; clock drawing test; Controlled Oral Word Association Test; Stroop test [41]; Trail Making Test A and B [43]; Frontal Assessment Battery [48] Subjective Memory Complaints Questionnaire [25] Unified Parkinson’s Disease Rating Scale [31] Geriatric Depression Scale [26] Blessed Dementia Scale-Activities of Daily Livings [27] Clinical Dementia Rating scale Global Deterioration Scale Neuropsychiatric Inventory [28]. Mini Nutritional Assessment [35] Pittsburgh Sleep Quality Index [36] Brain MRI Amyloid PET (^11^C-PiB PET in the KBASE and ^11^C-PiB PET or ^18^F-flutemetamol PET in the KBASE-V)	Cognitive Complaint Interview [23] Memory Assessment Clinics Questionnaire [24] Timed Up and Go Test [29] One leg standing [30] Simple appetite questionnaire [33] Eating Behavior Scale [34] International Physical Activity Questionnaire [32] Stanford Sleepiness Scale [37] Epworth Sleepiness Scale [38] Cardiovascular Health Study Frailty Index [39] CSF analysis of Aβ42, p-tau, and t-tau

AD, Alzheimer’s Disease; KBASE, Korean Brain Aging Study for the Early Diagnosis and Prediction of AD; KBASE-V, an independent validation cohort of KBASE; CSF, Cerebrospinal fluid.

**Table 2 jcm-08-00341-t002:** Baseline characteristics of the KBASE and KBASE-V participants.

Variables (Range)	CU of KBASE	CU of KBASE-V	MCI of KBASE	MCI of KBASE-V	ADD of KBASE	ADD of KBASE-V
*n*	291	167	139	72	87	56
Age, years	69.2 (8.1)	67.7 (7.8)	73.7 (7.0)	73.1 (8.4)	73.0 (8.1)	75.6 (7.5)
Female	149 (51%)	103 (62%) ^a^	92 (66%)	33 (46%) ^a^	60 (80%)	44 (79%)
Education, years	11.8 (4.8)	10.6 (5.1) ^b^	10.0 (4.5)	8.9 (4.2)	9.1 (5.4)	5.7 (4.1) ^b^
APOE ε4 carrier	53 (18%)	35 (21%)	47 (34%)	16 (22%)	52 (60%)	26 (46%)
MMSE (0–30)	26.9 (2.6)	26.5 (3.0)	22.5 (3.2)	22.7 (4.3)	16.4 (4.2)	16.0 (3.7)
CDR 0:0.5:1 ^§^	291:0:0	167:0:0	0:139:0	0:72:0	0:29:58	0:19:37
CDR-SB (0–18) ^§^	0.01 (0.06)	0.01 (0.07)	1.50 (0.65)	1.16 (0.86) ^†^	5.05 (1.45)	4.81 (2.03)
SBT (0–28) ^§^	2.5 (3.3)	3.1 (4.4)	9.1 (6.3)	8.6 (6.7)	18.9 (5.3)	19.3 (6.4)
CERAD, Z score	0.35 (0.56)	0.33 (0.47)	−0.98 (0.63)	−0.73 (0.59) ^b^	−1.60 (0.71)	−1.59 (0.69)
SMCQ (0–14) ^§^	2.1 (2.0)	3.9 (3.4) *	5.6 (3.1)	6.0 (3.6)	5.7 (3.0)	8.8 (3.8) ^‡^
GDS (0–30) ^§^	4.7 (4.9)	8.4 (6.5) *	10.3 (7.0)	10.4 (6.9)	7.9 (5.5)	11.9 (7.9) ^‡^
LM immediate recall	19.1 (7.3)	17.0 (8.1) *	9.2 (5.7)	7.7 (5.3) ^†^	3.8 (5.0)	2.5 (2.9)
LM delayed recall	13.9 (7.4)	12.0 (7.8) *	3.9 (4.6)	3.4 (4.0)	0.7 (4.3)	0.5 (1.3)
RCFT copy (0–36)	29.6 (6.2)	31.1 (6.6) *	24.7 (8.4)	27.1 (9.5)	17.8 (11.5)	14.8 (11.0)
Clock drawing (0–30)	27.7 (2.8)	27.6 (3.2)	25.6 (4.9)	24.8 (5.8)	20.9 (7.4)	17.2 (7.9)
Stroop test, Z score	0.18 (0.98)	0.12 (1.00)	−0.66 (0.98)	−0.55 (1.05)	−1.26 (1.15)	−1.34 (1.07)
COWAT	25.6 (11.4)	23.6 (10.4)	19.3 (9.0)	16.6 (9.5)	16.3 (10.0)	8.0 (7.8) ^‡^
A+	52 (18.1%) (*n* = 288)	26 (19.8%) (*n* = 131)	71 (53.0%) (*n* = 134)	28 (48.3%) (*n* = 58)	66 (81.5%) (*n* = 81)	36 (90.0%) (*n* = 40)

Data are shown as the mean (SD) or number (%). CU, cognitively unimpaired; MCI, mild cognitive impairment; ADD, Alzheimer’s disease dementia; KBASE, Korean Brain Aging Study for the Early Diagnosis and Prediction of AD; KBASE-V, an independent validation cohort of KBASE; APOE, apolipoprotein; MMSE, Mini-Mental State Examination; CDR, Clinical Dementia Rating scale; CDR-SB, CDR-Sum of Boxes; SBT, Short Blessed Test; CERAD, Consortium to Establish a Registry for AD; SMCQ, Subjective Memory Complaints Questionnaire; GDS, Geriatric Depression Scale; LM, Logical memory; RCFT, Rey Complex Figure Test; COWAT, Controlled Oral Word Association Test; A+, presence of amyloid pathology as determined by abnormal amyloid PET finding in the KBASE and by abnormal amyloid PET finding or CSF Aβ42 level below cut point in the KBASE-V. ^a^
*p* < 0.05 vs. its corresponding group of KBASE by chi-square test; ^b^
*p* < 0.05 vs. its corresponding group of KBASE by student *t*-test; * *p* < 0.05 vs. KBASE CU group from analysis of covariance (ANCOVA) with sex and education level as covariates; ^†^
*p* < 0.05 vs. KBASE MCI group from ANCOVA with sex as a covariate; ^‡^
*p* < 0.05 vs. KBASE ADD group from ANCOVA with education level as a covariate; ^§^ Increases in scores represent worsening.

**Table 3 jcm-08-00341-t003:** Demographic and clinical characteristics according to the Alzheimer’s disease (AD) clinical spectrum.

Participant Characteristics	CN (*n* = 71)	SCD (*n* = 96)	MCI (*n* = 72)	ADD (*n* = 56)	*P* *	*P* < 0.05 ^†^
Age, years	66.2 (7.8)	68.8 (7.7)	73.1 (8.4)	75.6 (7.5)	<0.001	b, c, d, e
Female	49 (69.0%)	54 (56.3%)	33 (45.8%)	44 (78.6%)	0.001	
Education, years	13.1 (4.0)	8.8 (5.2)	8.9 (4.2)	5.7 (4.1)	<0.001	a, b, c, e, f
Hypertension	28 (39.4%)	55 (57.3%)	42 (58.3%)	23 (41.1%)	0.029	
Diabetes Mellitus	19 (26.8%)	19 (19.8%)	20 (27.8%)	11 (19.5%)	0.502	
Dyslipidemia	57 (80.3%)	71 (74.0%)	55 (76.4%)	37 (66.1%)	0.321	
Current smoker	2 (2.8%)	3 (3.1%)	4 (5.6%)	1 (1.8%)	0.668	
Current drinker	27 (38.0%)	32 (33.3%)	18 (25.0%)	6 (10.7%)	0.004	
BMI, kg/m^2^	24.4 (2.4)	24.4 (3.0)	24.3 (2.8)	22.6 (3.0)	0.003	c, e, f
ASMI, kg/m^2^	6.7 (0.9)	7.3 (1.2)	7.2 (1.1)	6.6 (1.1)	0.013	
Hand grip, kg	30. 9 (7.9)	27.8 (8.7)	27.5 (8.8)	22.1 (7.2)	0.268	
Gait speed, m/s	1.18 (0.42)	1.04 (0.35)	1.02 (0.40)	0.86 (0.38)	0.676	
Sarcopenia	4 (5.6%)	8 (8.3%)	8 (11.1%)	16 (28.6%)	0.001	
MNA (range, 0–30)	25.4 (2.0)	24.4 (2.7)	24.4 (2.3)	21.9 (3.0)	<0.001	c, e, f
Meeting the PA guideline ^‡^	63 (88.7%)	66 (68.8%)	56 (77.8%)	36 (64.3%)	0.005	
Total sleep time, h	6.4 (1.4)	6.5 (1.3)	6.6 (1.7)	7.1 (2.0)	0.077	
ESS (range, 7–21) ^§^	4.5 (2.3)	4.7 (3.6)	3.1 (3.1)	3.7 (3.9)	0.010	b, d
APOE ε4 carrier	15 (21.1%)	20 (20.8%)	16 (22.2%)	26 (46.4%)	0.002	
Family history of dementia	11 (15.5%)	28 (29.2%)	16 (22.2%)	6 (10.7%)	0.028	
MMSE (range, 0–30)	28.1 (1.7)	25.3 (3.2)	22.7 (4.3)	16.0 (3.7)	<0.001	b, c, d, e, f
SBT (range, 0–28) ^§^	1.2 (1.5)	4.5 (5.3)	8.6 (6.7)	19.3 (6.4)	<0.001	b, c, d, e, f
CERAD, Z score	0.55 (0.43)	0.17 (0.43)	−0.73 (0.59)	−1.59 (0.69)	<0.001	a, b, c, d, e, f
CCI (range, 0–10) ^§^	1.6 (1.6)	3.3 (2.4)	3.8 (3.2)	6.2 (3.5)	<0.001	a, b, c, e, f
SMCQ (range, 0–14) ^§^	2.1 (2.0)	5.3 (3.6)	6.0 (3.6)	8.8 (3.8)	<0.001	a, b, c, e, f
MAC-Q (range, 6–30) ^§^	21.1 (2.6)	23.7 (4.2)	24.9 (3.8)	26.9 (4.2)	<0.001	a, b, c, e
GDS (range, 0–30) ^§^	6.3 (5.3)	10.0 (6.9)	10.4 (6.9)	11.9 (7.9)	0.001	a, b, c
NPI (range, 0–144) ^§^	0.2 (1.0)	1.6 (3.9)	2.5 (7.5)	13.7 (16.6)	<0.001	c, e, f
BDS-ADL (range, 0–17) ^§^	0.01 (0.12)	0.42 (0.70)	0.68 (0.82)	3.13 (2.41)	<0.001	c, e, f
CDR 0:0.5:1	71:0:0	96:0:0	0:72:0	0:19:37	<0.001	
CDR-SB (range, 0–18) ^§^	0.01 (0.06)	0.01 (0.07)	1.16 (0.86)	4.81 (2.03)	<0.001	b, c, d, e, f

Data are shown as the mean (SD) or number (%). CN, cognitively normal; SCD, subjective cognitive decline; MCI, mild cognitive impairment; ADD, Alzheimer’s disease dementia; BMI, body mass index; ASMI, appendicular skeletal muscle mass index; MNA, Mini Nutritional Assessment; PA, physical activity; ESS, Epworth Sleepiness Scale; APOE, apolipoprotein; MMSE, Mini-Mental State Examination; SBT, Short Blessed Test; CERAD, Consortium to Establish a Registry for AD; CCI, Cognitive Complaint Interview; SMCQ, Subjective Memory Complaints Questionnaire; MAC-Q, Memory Assessment Clinics Questionnaire; GDS, Geriatric Depression Scale; NPI, Neuropsychiatric Inventory; BDS-ADL, Blessed Dementia Scale-Activities of Daily Living; CDR, Clinical Dementia Rating scale; CDR-SB, CDR-Sum of Boxes; * Chi-square test for categorical variables, analysis of variance for age and education, analysis of covariance (ANCOVA) for Z score of CERAD with GDS as a covariate, ANCOVA for GDS as age, sex, and education level as covariates, and ANCOVA for other continuous variables with age, sex, education level, and GDS as covariates; ^†^ Tukey method for age and education and Bonferroni post hoc analysis for other continuous variables; a, CN vs. SCD; b, CN vs. MCI; c, CN vs. ADD; d, SCD vs. MCI; e, SCD vs. ADD; f, MCI vs. ADD; ^‡^ ≥600 metabolic equivalent minutes of physical activity (PA) per week; ^§^ Increases in scores represent worsening.

**Table 4 jcm-08-00341-t004:** Neuropsychological test results according to the AD clinical spectrum.

Neuropsychological Test (Range)	CN (*n* = 71)	SCD (*n* = 96)	MCI (*n* = 72)	ADD (*n* = 56)	*P* *	*P* < 0.05 ^†^
Auditory attention span						
Digit Span Forward (0–9)	6.7 (1.4)	6.1 (1.5)	5.8 (1.4)	5.5 (1.5)	0.748	
Digit Span Backward (0–8)	4.5 (1.2)	3.4 (1.3)	3.3 (1.1)	2.3 (1.3)	0.001	a, b, c
Language						
Verbal fluency, Z score,	0.49 (1.05)	−0.12 (1.05)	−0.52 (1.03)	−1.37 (0.97)	<0.001	a, b, c, e, f
Boston Naming, Z score	0.90 (0.71)	0.43 (0.79)	−0.10 (1.11)	−0.76 (1.35)	<0.001	a, b, c, d, e, f
Episodic memory						
Word list immediate recall, Z score	0.78 (0.81)	0.41 (0.88)	−0.71 (0.92)	−1.72 (1.03)	<0.001	b, c, d, e, f
Word list delayed recall, Z score	0.45 (0.77)	0.09 (0.79)	−1.29 (0.85)	−2.14 (0.82)	<0.001	b, c, d, e, f
Word list recognition, Z score	0.26 (1.01)	0.17 (0.66)	−1.26 (1.59)	−2.78 (1.55)	<0.001	b, c, d, e, f
Praxis recall, Z score	0.66 (0.82)	0.11 (0.81)	−0.86 (1.00)	−1.51 (0.75)	<0.001	a, b, c, d, e, f
RCFT, 3 min delayed recall (0–36)	16.89 (6.33)	12.89 (7.30)	8.80 (6.30)	1.20 (2.85)	<0.001	b, c, d, e, f
RCFT, 30 min delayed recall (0–36)	16.97 (6.76)	12.28 (7.56)	7.83 (6.27)	1.31 (2.32)	<0.001	a, b, c, d, e, f
LM immediate recall (0–50)	21.6 (7.1)	13.6 (7.1)	7.7 (5.3)	2.5 (2.9)	<0.001	a, b, c, d, e, f
LM delayed recall (0–50)	15.8 (7.3)	9.1 (6.9)	3.4 (4.0)	0.5 (1.3)	<0.001	a, b, c, d, e
LM recognition (0–30)	23.0 (3.6)	18.8 (4.2)	15.4 (5.5)	11.1 (6.5)	<0.001	b, c, d, e
Visuoconstruction						
Constructional praxis, Z score	0.34 (0.57)	0.09 (0.85)	−0.33 (1.29)	−0.86 (1.44)	<0.001	b, c, e, f
RCFT copy (0–36)	30.00 (0.96)	29.07 (0.47)	27.80 (0.86)	18.82 (1.04)	<0.001	c, e, f
Clock drawing (0–30)	28.8 (1.7)	26.8 (3.8)	24.8 (5.8)	17.2 (7.9)	<0.001	b, c, e, f
Block design (0–66)	34.3 (9.5)	26.4 (10.5)	24.8 (10.7)	12.3 (9.8)	<0.001	c, e, f
Executive function						
Stroop color reading, Z score	0.53 (0.91)	−0.19 (0.95)	−0.55 (1.05)	−1.34 (1.07)	<0.001	a, b, c, e, f
COWAT	28.4 (8.8)	20.0 (10.1)	16.6 (9.5)	8.0 (7.8)	<0.001	b, c, e, f
FAB (0–18)	15.7 (1.9)	13.9 (3.0)	11.4 (3.8)	7.9 (3.7)	<0.001	b, c, d, e, f
TMT-A (0–360 s)	50.8 (17.8)	67.0 (35.3)	93.3 (55.4)	168.9 (87.4)	<0.001	c, d, e, f
TMT-B (0–360 s)	125.9 (64.1)	171.0 (74.6)	190.1 (91.3)	255.8 (102)	<0.001	a, b, c, e, f

Data are shown as the mean (SD). CN, cognitively normal; SCD, subjective cognitive decline; MCI, mild cognitive impairment; ADD, Alzheimer’s disease dementia; RCFT, Rey Complex Figure Test; LM, Logical memory: COWAT, Controlled Oral Word Association Test; FAB, Frontal Assessment Battery; TMT, Trail making test; * Analysis of covariance (ANCOVA) for Z scores with GDS as a covariate and ANCOVA for raw scores with age, sex, education level, and GDS as covariates; ^†^ Bonferroni post hoc analysis; a, CN vs. SCD; b, CN vs. MCI; c, CN vs. ADD; d, SCD vs. MCI; e, SCD vs. ADD; f, MCI vs. ADD.

**Table 5 jcm-08-00341-t005:** Results of AD biomarkers according to the AD clinical spectrum.

	CN (*n* = 71)	SCD (*n* = 96)	MCI (*n* = 72)	ADD (*n* = 56)	*P* ^†^	*P* < 0.05 ^‡^
A+:A−	4 (8.9%):41	22 (25.6%):64	28 (48.3%):30	36 (90.0%):4	<0.001	
T+:T−	2 (6.9%):27	12 (20.7%):46	14 (48.3%):15	18 (78.3%):5	<0.001	
N+:N−	22 (31.0%):49	72 (75.0%):24	57 (79.2%):15	53 (94.7%):3	<0.001	
Amyloid PET, *n*	44	78	56	38 *		
Aβ deposition on PET	2 (4.5%)	18 (23.1%)	23 (41.1%)	32 (84.2%)	<0.001	
^18^F-flutemetamol PET, *n*	30	43	38	24		
FMM composite SUVR	0.58 (0.03)	0.62 (0.12)	0.67 (0.12)	0.81 (0.14)	<0.001	b, c, e, f
^11^C-PiB PET, *n*	14	35	18	13		
PiB composite SUVR	1.08 (0.04)	1.14 (0.16)	1.25 (0.28)	1.56 (0.41)	<0.001	c, e, f
Cerebrospinal fluid, *n*	29	58	29	23		
Aβ42, pg/mL	548.6 (87.1)	510.1 (98.5)	415.3 (141.4)	306.2 (126.3)	<0.001	b, c, d, e, f
T-tau, pg/mL	49.6 (10.6)	52.3 (21.4)	55.4 (18.8)	91.1 (62.0)	<0.001	c, e, f
P-tau, pg/mL	16.8 (5.0)	18.5 (7.4)	23.6 (14.7)	37.0 (24.1)	<0.001	c, e, f
T-tau/Aβ42	0.09 (0.02)	0.12 (0.11)	0.16 (0.10)	0.37 (0.34)	<0.001	c, e, f
P-Tau/Aβ42	0.03 (0.01)	0.04 (0.03)	0.07 (0.07)	0.15 (0.11)	<0.001	b, c, e, f
Brain MRI, *n*	71	96	72	56		
Cortical thickness, mm	3.17 (0.09)	3.03 (0.15)	3.05 (0.14)	2.97 (0.15)	<0.001	a, b, c, f
Hippocampal volume cm^3^	5.12 (0.63)	5.09 (0.76)	4.76 (0.77)	4.40 (0.80)	<0.001	b, c, d, e, f

Data are shown as the mean (SD) or number (%). CN, cognitively normal; SCD, subjective cognitive decline; MCI, mild cognitive impairment; ADD, Alzheimer’s disease dementia; A−, absence of amyloid pathology determined by normal amyloid PET finding or CSF Aβ42 level above cut point; A+, presence of amyloid pathology determined by abnormal amyloid PET finding or CSF Aβ42 level below cut point; T−, normal CSF p-tau level below cut point; T+, abnormal CSF p-tau level above cut point; N−, absence of neurodegeneration or neuronal injury determined by cortical thickness or hippocampal volume; N+, presence of neurodegeneration or neuronal injury determined by cortical thickness or hippocampal volume; FMM, flutemetamol; SUVR, standardized uptake value ratio; *n*, number; Aβ, amyloid beta; t-tau, total tau; p-tau, phosphorylated tau; * A patient with amyloid deposition on historical ^18^F-florbetapir PET was included; ^†^ Chi-square test for categorical variables, analysis of variance for continuous variables; ^‡^ Tukey method; a, CN vs. SCD; b, CN vs. MCI; c, CN vs. ADD; d, SCD vs. MCI; e, SCD vs. ADD; f, MCI vs. ADD.

**Table 6 jcm-08-00341-t006:** ATN classification in regard to the biomarker profile and cognitive stage among participants with cerebrospinal fluid (CSF) AD biomarker results.

ATN Profile	Biomarker Category	Syndromal Cognitive Stage			
		CN (*n* = 29)	SCD (*n* = 58)	MCI (*n* = 29)	ADD (*n* = 23)
A−/T−/N−	Normal AD biomarkers	20 (69.0%)	13 (22.4%)	2 (6.9%)	0
A+/T−/N−	Alzheimer’s pathologic change	1 (3.4%)	1 (1.7%)	3 (10.3%)	0
A+/T+/N−	Alzheimer’s disease	0	3 (5.2%)	2 (6.9%)	1 (4.4%)
A+/T+/N+	Alzheimer’s disease	0	6 (10.3%)	7 (24.1%)	17 (73.9%)
A+/T−/N+	Alzheimer’s and SNAP	2 (6.9%)	4 (6.9%)	4 (13.8%)	3 (13.0%)
A−/T+/N−	Non-AD pathologic change	2 (6.9%)	0	2 (6.9%)	0
A−/T+/N+	Non-AD pathologic change	0	3 (5.2%)	3 (10.3%)	0
A−/T−/N+	Non-AD pathologic change	4 (13.8%)	28 (48.3%)	6 (20.7%)	2 (8.7%)

Data are shown as number (%). CN, cognitively normal; SCD, subjective cognitive decline; MCI, mild cognitive impairment; ADD, Alzheimer’s disease dementia; ATN, amyloid, tau, and neurodegeneration or neuronal injury; A−, absence of amyloid pathology determined by normal amyloid PET finding or CSF Aβ42 level above cut point; A+, presence of amyloid pathology determined by abnormal amyloid PET finding or CSF Aβ42 level below cut point; T−, normal CSF p-tau level below cut point; T+, abnormal CSF p-tau above cut point; N−, absence of neurodegeneration or neuronal injury determined by cortical thickness or hippocampal volume; N+, presence of neurodegeneration or neuronal injury determined by cortical thickness or hippocampal volume; SNAP, suspected non Alzheimer’s pathologic change.

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
