# Peer review of "Clinical and Biomarker Characteristics According to Clinical Spectrum of Alzheimer’s Disease (AD) in the Validation Cohort of Korean Brain Aging Study for the Early Diagnosis and Prediction of AD"

_jcm, 2019, doi:10.3390/jcm8030341_

Reviewer 1 Report

This paper describes the KBASE validation cohort for Alzheimer's disease. The manuscript is well-written and the content will be of interest to the JCM readership. I note a few minor limitations with suggestions for improvement. I also note some required major changes.

Major changes:

M1. The first major change involves the manuscript focus. It currently presents descriptive statistics and group comparisons within the KBASE-V cohort. While these are interesting and should be presented, each clinical group should be compared with its corresponding group in the main KBASE cohort. This is based on my presumption that KBASE-V is for validation purposes against KBASE. If my presumption is correct, then it is important to report on the statistical matching between KBASE-V and KBASE.

M2.1 The second major change is to report on corrections for multiple comparisons. The reader is bombarded with p-values, but there is no mention of correcting for multiple comparisons. This may require the authors to perform the statistical analyses over again (in particular the null hypothesis tests for group comparisons) with adequate corrections for multiple comparisons. I think that only one of the reported post hoc tests (Tukey's method) automatically corrects for multiple comparisons.

M2.2 The authors report that they calculate Z scores for the neuropsych tests based on "the age-, sex- and education-matched control group". However, they go on to say that the controls were significantly younger than the MCI and ADD groups. Which is it? Are the groups matched, or are they significantly different? This conflict needs to be resolved.

Minor changes:

m1. The apparent statistical difference in age between controls and MCI/ADD is a confounder in any disease-related analysis that incorporates group comparisons. Age is the primary risk factor for Alzheimer's disease, and so the groups should match in order to draw sensible conclusions.

m2. The reader would like to see a graphical representation/visualisation of the A/T/N data in Table 4. Perhaps a stacked bar graph showing proportions of diagnostic groups within each A/T/N category.

m3. Line 393 (page 11): MMSE in the SCD group may be worse than in controls due to higher depression in SCD (Lines 401-402). The authors should investigate (and report in a revised manuscript) whether depression can explain the MMSE differences between the SCD and control groups.

m4. Line 406 (page 11): Surely this is unremarkable. Circular at best. My understanding is that subjective cognitive decline is measured by questionnaire(s), so it's unremarkable that a subjective memory questionnaire is useful for separating the CN and SCD groups. It's like saying "AD diagnostic criteria is useful for distinguishing ADD from not-ADD". The unremarkable/circular statement should be removed (or qualified explicitly as being true by definition). The rest of the paragraph is interesting.

m5. Line 412: The authors comment that delayed recall LM scores were not different between MCI and ADD groups. They then draw the conclusion that LM may not be suitable for assessing MCI-to-ADD. This conclusion would be supported by an added comment about whether this finding has also been made elsewhere, such as in the ADNI cohort.

m6. Minor typos including Stoop (Stroop), et al.

Author Response

Response to Reviewer 1 Comments

This paper describes the KBASE validation cohort for Alzheimer's disease. The manuscript is well-written and the content will be of interest to the JCM readership. I note a few minor limitations with suggestions for improvement. I also note some required major changes.

Major changes:

Point M1. The first major change involves the manuscript focus. It currently presents descriptive statistics and group comparisons within the KBASE-V cohort. While these are interesting and should be presented, each clinical group should be compared with its corresponding group in the main KBASE cohort. This is based on my presumption that KBASE-V is for validation purposes against KBASE. If my presumption is correct, then it is important to report on the statistical matching between KBASE-V and KBASE.

Response 1: I agree with you. Base on your comments, I added the statistical comparisons between each clinical group and its corresponding group in the main KBASE cohort. I added Table 2 and modified the manuscript as follows.

I added “To compare the continuous clinical variables between each group of KBASE-V and its corresponding group of KBASE, ANCOVA tests were conducted with sex and education level as covariates in the CU group, sex as a covariate in the MCI group, and education level as a covariate in the ADD group.” on lines 273-276 of page 6 in statistical analysis. 

I added Table 2 and “There was no significant difference in age, MMSE, SBT, CDR, Stroop test, clock drawing, and the prevalence of APOE ε4 carriers and A+ between each clinical group and its corresponding group in the KBASE (Table 2). The education level in the CU and ADD groups of KBASE-V was lower than the corresponding group of KBASE. In comparison to the corresponding group of KBASE, the prevalence of women was higher in the KBASE-V CU group and lower in the KBASE-V MCI group. The MCI group of KBASE-V had higher CERAD and lower CDR-SB scores than the MCI group of KBASE. The CU and ADD groups of KBASE-V showed higher GDS and SMCQ scores than the corresponding group in the KBASE.” on lines 286-293 of page 7.  

Table 2. Baseline characteristics of the KBASE and KBASE-V participants

Variables

(range)

CU of KBASE

CU of

KBASE-V

MCI of KBASE

MCI of

KBASE-V

ADD of

KBASE

ADD of

KBASE-V

n

291

167

139

72

87

56

Age,   years

69.2   (8.1)

67.7   (7.8)

73.7   (7.0)

73.1   (8.4)

73.0   (8.1)

75.6   (7.5)

Female

149   (51%)

103   (62%)a

92   (66%)

33   (46%)a

60   (80%)

44   (79%)

Education,   years

11.8   (4.8)

10.6   (5.1)b

10.0   (4.5)

8.9   (4.2)

9.1   (5.4)

5.7   (4.1)b

APOE   ε4 carrier

53   (18%)

35   (21%)

47   (34%)

16   (22%)

52   (60%)

26   (46%)

MMSE   (0-30)

26.9   (2.6)

26.5   (3.0)

22.5   (3.2)

22.7   (4.3)

16.4   (4.2)

16.0   (3.7)

CDR 0:   0.5: 1§

291 :   0: 0

167:   0: 0

0:   139: 0

0: 72:   0

0: 29:   58

0: 19:   37

CDR-SB   (0-18)§

0.01   (0.06)

0.01   (0.07)

1.50   (0.65)

1.16   (0.86)

5.05   (1.45)

4.81   (2.03)

SBT   (0-28)§

2.5   (3.3)

3.1   (4.4)

9.1   (6.3)

8.6   (6.7)

18.9   (5.3)

19.3   (6.4)

CERAD,   Z score

0.35   (0.56)

0.33   (0.47)

-0.98 (0.63)

-0.73   (0.59)b

-1.60   (0.71)

-1.59   (0.69)

SMCQ   (0-14)§

2.1   (2.0)

3.9   (3.4)*

5.6   (3.1)

6.0   (3.6)

5.7   (3.0)

8.8   (3.8)

GDS   (0-30)§

4.7   (4.9)

8.4   (6.5)*

10.3   (7.0)

10.4   (6.9)

7.9   (5.5)

11.9   (7.9)

LM   immediate recall

19.1   (7.3)

17.0   (8.1)*

9.2   (5.7)

7.7   (5.3)

3.8   (5.0)

2.5   (2.9)

LM   delayed recall

13.9   (7.4)

12.0   (7.8)*

3.9   (4.6)

3.4   (4.0)

0.7   (4.3)

0.5   (1.3)

RCFT   copy (0-36)

29.6   (6.2)

31.1   (6.6)*

24.7   (8.4)

27.1   (9.5)

17.8   (11.5)

14.8   (11.0)

Clock   drawing (0-30)

27.7   (2.8)

27.6   (3.2)

25.6   (4.9)

24.8   (5.8)

20.9   (7.4)

17.2   (7.9)

Stroop   test, Z score

0.18   (0.98)

0.12   (1.00)

-0.66   (0.98)

-0.55   (1.05)

-1.26   (1.15)

-1.34   (1.07)

COWAT

25.6   (11.4)

23.6   (10.4)

19.3   (9.0)

16.6   (9.5)

16.3   (10.0)

8.0   (7.8)

A+

52   (18.1%) (n=288)

26   (19.8%)

(n=131)

71 (53.0%)

(n=134)

28   (48.3%)

(n=58)

66   (81.5%)

(n=81)

36   (90.0%)

(n-40)

Data are shown as the mean (SD) or number (%). CU, cognitively unimpaired; MCI, mild cognitive impairment; ADD, Alzheimer’s disease dementia; KBASE, Korean Brain Aging Study for the Early Diagnosis and Prediction of AD; KBASE-V, an independent validation cohort of KBASE; APOE, apolipoprotein; MMSE, Mini-Mental State Examination; CDR, Clinical Dementia Rating scale; CDR-SB, CDR-Sum of Boxes; SBT, Short Blessed Test; CERAD, Consortium to Establish a Registry for AD; SMCQ, Subjective Memory Complaints Questionnaire; GDS, Geriatric Depression Scale; LM, Logical memory; RCFT, Rey Complex Figure Test; COWAT, Controlled Oral Word Association Test; A+, presence of amyloid pathology determined by abnormal amyloid PET finding in the KBASE and by abnormal amyloid PET finding or CSF Aβ42 level below cut point in the KBASE-V. a, p < 0.05 vs. its corresponding group of KBASE by chi-square test; b, p < 0.05 vs. its corresponding group of KBASE by student t-test; *p < 0.05 vs. KBASE CU group from analysis of covariance (ANCOVA) with sex and education level as covariates; p < 0.05 vs. KBASE MCI group from ANCOVA with sex as a covariate; p < 0.05 vs. KBASE ADD group from ANCOVA with education level as a covariate; §Increases in scores represent worsening.

On lines 431-449 of page 13 in the discussion section, I changed “The KBASE-V is a suitable cohort to validate potential AD biomarkers and to identify risk and prognostic factors of AD. A total of 216 (73.2%) participants underwent amyloid PET. The presence or absence of amyloid pathology according to CSF analysis or amyloid PET was confirmed in 229 (77.6%) participants. The percentage of participants in KBASE-V for whom amyloid biomarkers were examined was not less than in the previous cohort [49,65]”  into  “Each clinical group of KBASE-V showed generally comparable baseline characteristics with its corresponding group of KBASE, with some differences. General cognitive function such as MMSE, SBT, and CDR in each clinical group was similar between the KBASE and KBAE-V. Age and the prevalence of APOE ε4 carriers were also similar between each clinical group of KBASE-V and its corresponding group of KBASE. The education level in the CU and ADD groups of KBASE-V was lower than the corresponding group of KBASE. The cause is that the KBASE-V is a nationwide cohort and the KBASE is a cohort of highly educated people living in the capital and surrounding areas. A higher GDS score in the CU and ADD groups of KBASE-V may also be related to the socioeconomic differences between the KBASE and KBAE-V. The CERAD score in the CU and ADD groups of KBASE-V was similar to that in the corresponding group of KBASE. The cognitive function of the MCI group of KBASE-V appears to be slightly better than the corresponding group of KBASE, as the MCI participants in KBASE-V showed higher CREAD and lower CDR-SB scores than the MCI group in KBASE. The prevalence of A+ was similar between each clinical group of KBASE-V and its corresponding group of KBASE. A total of 216 (73.2%) participants underwent amyloid PET in the KBASE-V. The presence or absence of amyloid pathology according to CSF analysis or amyloid PET was confirmed in 229 (77.6%) participants of the KBASE-V. The percentage of participants in KBASE-V for whom amyloid biomarkers were examined was not less than in the previous cohort [49,65]. The KBASE-V is a suitable cohort for reconfirming potential new AD biomarkers such as blood biomarkers identified in the KBASE and identifying the risk and prognostic factors of AD.”.

Point M2.1 The second major change is to report on corrections for multiple comparisons. The reader is bombarded with p-values, but there is no mention of correcting for multiple comparisons. This may require the authors to perform the statistical analyses over again (in particular the null hypothesis tests for group comparisons) with adequate corrections for multiple comparisons. I think that only one of the reported post hoc tests (Tukey's method) automatically corrects for multiple comparisons.

Response M2.1: I agree with you. When a statistical significant overall difference was detected in the ANOVA test, pairwise comparisons on means between diagnosis groups were conducted by Tukey method. When a statistical significant overall differences was detected in ANCOVA test with age, sex, and education as covariates, pairwise comparisons on means between diagnosis groups were conducted by Least Significant Difference (LSD) method. Based on your comments, I changed LSD method into Bonferroni post hoc analysis for corrects for multiple comparisons. I modified in Tables 3 and 4 based on the new analysis and amended the manuscript as follows.

On lines 266-273 of page 6, I changed “We used the Tukey method for post hoc analysis. Analysis of covariance was performed for statistical analysis of the MMSE, SBT, and raw scores of neuropsychological tests with age, sex, and educational years as covariates. The Least Significant Difference method was used for post hoc analysis” into  “When a statistical significant overall difference was detected in the ANOVA test, pairwise comparisons on means between diagnosis groups were conducted by Tukey method. Analysis of covariance (ANCOVA) was performed for statistical analysis of BMI, ASMI, hand grip, gait speed, MNA, total sleep time, ESS, CCI, MMSE, SBT, SMCQ, MAC-Q, GDS, NPI, BDS-ADL, CDR-SB, and raw scores of neuropsychological tests with age, sex, and education level as covariates. When a statistical significant overall difference was detected in the ANCOVA test, pairwise comparisons on means between diagnosis groups were conducted by Bonferroni post hoc analysis for correction of multiple comparisons.”. 

On lines 315-317 of page 8, I changed “The MMSE and CERAD scores differed by diagnostic group with better performance….” into “The MMSE, SBT, and CDR-SB scores were significantly different between all groups, except between the CN and SCD groups (Figure 1A). The CERAD scores differed by diagnostic group with better performance … ”.

In the legend of Figure 1 (lines 338-342) of page 8, I changed “The Mini-Mental State Examination (MMSE) and Consortium to Establish a Registry for AD (CERAD) Z scores were significantly different between all groups…” into “In the Mini-Mental State Examination (MMSE) scores, CSF Aβ42 level and hippocampal volume, there was a significant difference between all groups, except between cognitively normal (CN) and subjective cognitive decline (SCD) groups (A, E, and F). The Consortium to Establish a Registry for AD (CERAD) Z scores were significantly different between all groups…”, and I modified Figure 1(A).

Point M2.2 The authors report that they calculate Z scores for the neuropsychological tests based on "the age-, sex- and education-matched control group". However, they go on to say that the controls were significantly younger than the MCI and ADD groups. Which is it? Are the groups matched, or are they significantly different? This conflict needs to be resolved.

Response M2.1: I am sorry to confuse you with my ambiguous expression. I changed “the age-, sex- and education-matched control group” into “the age-, sex- and education-matched normal population” on lines 264-265 of page 6.

Minor changes:

Point m1. The apparent statistical difference in age between controls and MCI/ADD is a confounder in any disease-related analysis that incorporates group comparisons. Age is the primary risk factor for Alzheimer's disease, and so the groups should match in order to draw sensible conclusions.

Response m1: I agree with you. Based on your comments, ANCOVA tests for clinical continuous variables in Table 3 were conducted with age, sex, and education level as covariates. I already had conducted the ANCOVA tests with age, sex, and education level as covariates for raw scores of neuropsychological tests in Table 4.

Based on your comments, I modified the manuscript as follows.

On lines 267-273 of page 6, I changed “Analysis of covariance was performed for statistical analysis of the MMSE, SBT, and raw scores of neuropsychological tests with age, sex, and educational years as covariates. The Least Significant Difference method was used for post hoc analysis” into  “Analysis of covariance (ANCOVA) was performed for statistical analysis of BMI, ASMI, hand grip, gait speed, MNA, total sleep time, ESS, CCI, MMSE, SBT, SMCQ, MAC-Q, GDS, NPI, BDS-ADL, CDR-SB, and raw scores of neuropsychological tests with age, sex, and education level as covariates. When a statistical significant overall difference was detected in the ANCOVA test, pairwise comparisons on means between diagnosis groups were conducted by Bonferroni post hoc analysis for correction of multiple comparisons.”. 

 I modified in Tables 3 and some results section based on the new analysis.

Point m2. The reader would like to see a graphical representation/visualisation of the A/T/N data in Table 4. Perhaps a stacked bar graph showing proportions of diagnostic groups within each A/T/N category.

Response m2: I agree with you. Based on your comments, I added Figure 2 of a stacked bar graph showing proportions of A/T/N category within each diagnostic group on page 12-13.

Figure 2. A stacked bar graph showing the proportions of ATN categories within each diagnostic group among the KBASE-V participants with CSF AD biomarker results. The proportion of participants in the Alzheimer's continuum (A+/T+/N+, A+/T+/N-, A+/T-/N+, and A+/T-/N-) was 10.3% in the CN group, 24.1% in the SCD group, 55.1% in the MCI group, and 91.2% in the ADD group. CN, cognitively normal; SCD, subjective cognitive decline; MCI, mild cognitive impairment; ADD, Alzheimer’s disease dementia; ATN, amyloid, tau, and neurodegeneration or neuronal injury; A−, absence of amyloid pathology determined by normal amyloid PET finding or CSF Aβ42 level above cut point; A+, presence of amyloid pathology determined by abnormal amyloid PET finding or CSF Aβ42 level below cut point; T−, normal CSF p-tau level below cut point; T+, abnormal CSF p-tau level above cut point; N−, absence of neurodegeneration or neuronal injury determined by cortical thickness or hippocampal volume; N+, presence of neurodegeneration or neuronal injury determined by cortical thickness or hippocampal volume.

Point m3. Line 393 (page 11): MMSE in the SCD group may be worse than in controls due to higher depression in SCD (Lines 401-402). The authors should investigate (and report in a revised manuscript) whether depression can explain the MMSE differences between the SCD and control groups.

Response m3: After Bonferroni post hoc analysis based on your comments (M2.1), there was no difference in MMSE score between the CN and SCD groups.

On lines 476-478 of page 14, I changed “General cognitive function was worse in the SCD group on the MMSE when compared with the CN group. These results were not compatible with those from previous studies reporting that MMSE scores were not different between CN and SCD groups [68,69]. This may be due to educational differences between the CN and SCD groups despite adjusting for education. Alternatively, this may be due to the very low prevalence of amyloid pathology in the CN group. In addition, there were no participants with AD but only participants with Alzheimer’s pathologic changes in the CN group according to ATN classification [14]”  into  “General cognitive function such as MMSE and CDR-SB was not different between the CN and SCD groups. These results were also compatible with those from previous studies [68,69].”.

Point m4. Line 406 (page 11): Surely this is unremarkable. Circular at best. My understanding is that subjective cognitive decline is measured by questionnaire(s), so it's unremarkable that a subjective memory questionnaire is useful for separating the CN and SCD groups. It's like saying "AD diagnostic criteria is useful for distinguishing ADD from not-ADD". The unremarkable/circular statement should be removed (or qualified explicitly as being true by definition). The rest of the paragraph is interesting.

Response m4: I agree with you. I removed “subjective memory questionnaires are useful to distinguish between CN and SCD” and changed “In this study, we also confirmed that subjective memory questionnaires are useful to distinguish between CN and SCD. However, the questionnaires could not differentiate between the SCD and MCI groups” into “ In this study, the subjective memory questionnaires could not differentiate between the SCD and MCI groups.” on lines 483-484 of page 14.

Point m5. Line 412: The authors comment that delayed recall LM scores were not different between MCI and ADD groups. They then draw the conclusion that LM may not be suitable for assessing MCI-to-ADD. This conclusion would be supported by an added comment about whether this finding has also been made elsewhere, such as in the ADNI cohort.

Response m5: I agree with you. I modified the manuscript as “LM tests may not be suitable to monitor the natural course of ADD” as follows.

On lines 490-495 of page 14, I changed “Delayed recall scores of LM were not different between the MCI and ADD groups. LM delayed recall seems to show a floor effect even at the MCI stage. Therefore, LM tests may not be suitable to monitor the natural course of MCI and ADD”  into  “The delayed recall score of LM did not differ between the MCI group and ADD group after correction for multiple comparisons. The LM delayed recall test appears to show a floor effect even at the mild ADD stage, since the LM delayed recall score was almost zero in the participants with mild ADD of a CDR 0.5 or 1 in the KBASE-V. The results are similar to those of the previous report [73]. Therefore, LM tests may not be suitable to monitor the natural course of ADD.”.

I added “73. Lemos, R.; Cunha, C.; Marôco, J.; Afonso, A.; Simões, M.R.; Santana, I. Free and Cued Selective Reminding Test is superior to the Wechsler Memory Scale in discriminating mild cognitive impairment from Alzheimer's disease. Geriatr Gerontol Int 2015, 15, 961-968, doi:10.1111/ggi.12374. ” on lines 762-764 of page 19 of the reference section.

Point m6. Minor typos including Stoop (Stroop), et al.

Response m6: I corrected other typing errors as well as Stoop.

Reviewer 2 Report

Hwang and colleagues present a validation study in an independent cohort of CN, SCD, MCI, and ADD.  This is an important study because recent studies in the US have shown that the ATN scheme may not directly translate across racial groups.  The authors are also mindful of the importance for a validation cohort.  However, it was not clear what they were validating, as it seemed they were using KBASE-V to derive cut-off values.  This may not be the case, but the manuscript is not clear enough on what’s from KBASE and what’s from KBASE-V.

There are also other issues that need to be addressed, including:

1) CSF Abeta42 levels are much higher than typically reported, and may have to do with technical processing.  The authors need to address why their cutoff is twice that from autopsy-based studies.

2) There is circular reasoning in the definition of ATN, eg CSF Ab42 cut off was determined based on amyloid PET, but then A is the sum of abnormal PET and abnormal Ab42.  If KBASE had these values, it’s better to use those cutoffs.  Otherwise, they should use published cutoffs.

3) CSF tau is strongly correlated with ptau, so ptau can’t be used for T while ttau is used for N.

4) showing group level differences (are error bars SD or SEM?) misses the point of validation.  The authors must report how each subject is classified according to KBASE values, or use internal cross validation to have some objective assessment.

5) there is no intermediate precision reported for the CSF measures.

Author Response

Response to Reviewer 2 Comments

Point 1: Hwang and colleagues present a validation study in an independent cohort of CN, SCD, MCI, and ADD. This is an important study because recent studies in the US have shown that the ATN scheme may not directly translate across racial groups. The authors are also mindful of the importance for a validation cohort. However, it was not clear what they were validating, as it seemed they were using KBASE-V to derive cut-off values. This may not be the case, but the manuscript is not clear enough on what’s from KBASE and what’s from KBASE-V.

Response 1: I agree with you. The KBASE-V is an independent nationwide cohort to reconfirm new potential biomarkers, particularly blood biomarkers, identified in the KBASE study and is intended to find risk and prognostic factors for AD as additional studies. I added Table 1 that shows the same measures with the KBASE and the additional measures of the KBASE-V and modified the manuscript as follows, based on your comments. 

On lines 87-95 of page 2, I changed “The Korean Brain Aging Study for the Early Diagnosis and Prediction of AD (KBASE) is a prospective cohort study that began in September 2014 and is aimed at recruiting cognitively normal (CN) individuals and patients with MCI or ADD to identify new biomarkers and risk factors for AD [17]. The validation cohort of KBASE (KBASE-V) is an independent cohort used to validate the potential biomarkers identified in the KBASE study and find risk and prognostic factors for AD. The KBASE-V consisted of participants with ADD, MCI, or SCD as well as CN individuals without SCD”  into  “The Korean Brain Aging Study for the Early Diagnosis and Prediction of AD (KBASE) is a prospective cohort study that began in Seoul and surrounding areas in 2014 and is aimed at recruiting cognitively unimpaired (CU) individuals and patients with MCI or ADD to identify new biomarkers, particularly blood biomarkers, for the early detection of AD and risk factors for AD [17]. The validation cohort of KBASE (KBASE-V) is an independent nationwide cohort to reconfirm new potential biomarkers identified in the KBASE study and is intended to find risk and prognostic factors for AD as additional studies. The KBASE-V also consisted of participants with ADD, MCI, or CU, and CU individuals were divided into SCD and cognitively normal (CN) individuals without SCD.”.

I added “The same measures with the KBASE and the additional measures of the KBASE-V among those performed in the KBASE-V are shown in Table 1. The neuropsychological tests and clinical evaluations performed in the KBASE-V were basically the same as the KBASE and additional evaluations of lifestyle risk factors were made in the KBASE-V.” on lines 279-282 of page 6, and Table 1 on page 6-7. 

Table 1. The same measures with the KBASE and the additional measures of the KBASE-V

Same measures with the KBASE

Additional measures in the KBASE-V

Inclusion and exclusion   criteria

Mini-Mental State   Examination [18]

Consortium to Establish a Registry for AD [18]

Short   Blessed Test [42]

Detailed   neuropsychological tests: digit span forward and backward tests [44]; logical   Memory test [45]; Rey Complex Figure Test [46]; block design [47]; clock   drawing test; Controlled Oral Word Association Test; Stroop test [41]; Trail   Making Test A and B [43]; Frontal Assessment Battery [48]

Subjective Memory Complaints Questionnaire [25]

Unified Parkinson’s Disease   Rating Scale [31]

Geriatric Depression Scale   [26]

Blessed Dementia   Scale-Activities of Daily Livings [27]

Clinical Dementia Rating   scale

Global Deterioration Scale

Neuropsychiatric Inventory   [28].

Mini Nutritional Assessment   [35]

Pittsburgh Sleep Quality   Index [36]

Brain MRI

Amyloid PET (11C-PiB PET in the KBASE and

11C-PiB PET or 18F-flutemetamol PET in   the KBASE-V)

Cognitive Complaint Interview   [23]

Memory Assessment Clinics   Questionnaire [24]

Timed Up and Go Test [29]

One leg standing [30]

Simple appetite   questionnaire [33]

Eating Behavior Scale [34]

International Physical Activity Questionnaire [32]

Stanford Sleepiness Scale   [37]

Epworth Sleepiness Scale   [38]

Cardiovascular Health Study   Frailty Index [39]

CSF analysis of Aβ42,   p-tau, and t-tau

AD, Alzheimer’s Disease; KBASE, Korean Brain Aging Study for the Early Diagnosis and Prediction of AD; KBASE-V, an independent validation cohort of KBASE; CSF, Cerebrospinal fluid.

Point 2: CSF Abeta42 levels are much higher than typically reported, and may have to do with technical processing. The authors need to address why their cutoff is twice that from autopsy-based studies.

Response 2: I agree with your opinion. I addressed why CSF Abeta42 levels are much higher than typically reported in this study in the discussion.

On lines 500-512 of page 14, I added “The cutoff of CSF Aβ42 levels for distinguishing ADD from the CN participants in the KBASE-V was much higher than in the ADNI [65]. The possible causes of the discrepancy with the ADNI were as follows. First, it is well recognized that interlaboratory variability in immunoassay-based measurement of CSF Aβ42 is very large [76], hence the interlaboratory variability might be a reason of higher cut-off value in our study. Second, a small sample size may be related to the higher cutoff value of CSF Aβ42 levels. Third, it may be associated with age difference between the ADNI and KBASE-V. The participants in the KBASE-V were younger than those in the ADNI. Fourth, the mean score of CSF Aβ42 levels in each group of the KBASE-V was higher than in the corresponding group of ADNI, but was similar to that in the corresponding group of Japanese ADNI [65]. There may be ethnic difference in CSF Aβ42 levels. Finally, we could not exclude a technical error. The possibility seems to be low, since we determined acceptable precision of xMAP-Luminex with AlzBio3 kit (average between-run %CV for Aβ42 through 5 runs was 9.1 (2.0)%) using pooled 9 CSFs before measurement of our samples.”.

I changed “reference 76. Rizzi, L.; Maria Portal, M.; Batista, C.E.A.; Missiaggia, L.; Roriz-Cruz, M. CSF Aβ1-42, but not p-Tau181, differentiates aMCI from SCI. Brain Res 2018, 1678, 27-31, doi:10.1016/j.brainres.2017.10.008” into “reference 76. Dumurgier, J.; Vercruysse, O.; Paquet, C.; Bombois, S.; Chaulet, C.; Laplanche, J.L.; Peoc'h, K.; Schraen, S.; Pasquier, F.; Touchon, J., et al. Intersite variability of CSF Alzheimer's disease biomarkers in clinical setting. Alzheimers Dement 2013, 9, 406-13, doi: 10.1016/j.jalz.2012.06.006.” on lines 770-772 of page 19..

Point 3: There is circular reasoning in the definition of ATN, eg CSF Ab42 cut off was determined based on amyloid PET, but then A is the sum of abnormal PET and abnormal Ab42.  If KBASE had these values, it’s better to use those cutoffs.  Otherwise, they should use published cutoffs.

Response 3:  I agree with you. Based on your opinion, I evaluated the cutoff values of CSF Aβ42, t-tau and p-tau based on clinical diagnosis, and modified the manuscript as follows.

On lines 252-254 of page 6, I changed “The cutoff values that provided the highest Youden's index for classifying participants with amyloid deposition on PET from those without amyloid deposition on PET were 466.9 pg/ml (sensitivity 89.7%, specificity 83.9%) for Aβ42, 63.7 pg/ml (sensitivity 56.4%, specificity 88.4%) for t-tau, and 21.0 pg/ml (sensitivity 76.9, specificity 81.6%) for p-tau”  into  “The cutoff values that provided the highest Youden's index for ADD diagnosis from the CN group were 433.68 pg/ml (sensitivity 91.3%, specificity 89.7%) for Aβ42, 57.77 pg/ml (sensitivity 69.6%, specificity 82.8%) for t-tau, and 22.55 pg/ml (sensitivity 78.3%, specificity 93.1%) for p-tau.”.

I changed “(CSF Aβ42 levels ≤ 466.9 pg/mL….” into “(CSF Aβ42 levels ≤ 433.68  pg/mL….” on line 257 of page 6, and “CSF p-tau ≥ 21.0 pg/mL” into “CSF p-tau ≥ 22.55pg/mL” on line 258 of page 6.

I modified the frequency (%) of ATN in Tables 5 and 6 and based on the new analysis.

Point 4: CSF tau is strongly correlated with ptau, so p-tau can’t be used for T while t-tau is used for N.

Response 4: I agree with you. Based on your opinion, I have removed t-tau from the criteria for ‘N’, and modified the manuscript as follows.

On lines 259-260 of page 6, I changed “N+ refers to the neurodegeneration biomarker (hippocampal volume ≤ 4.67 cm3 or mean cortical thickness ≤ 3.07 mm or CSF t-tau ≥ 63.7 pg/mL)”  into “N+ refers to the neurodegeneration biomarker (hippocampal volume ≤ 4.67 cm3 or mean cortical thickness ≤ 3.07 mm )”

On lines 385-387 and 413-415 of page 10, I changed “N−, absence of neurodegeneration or neuronal injury determined by cortical thickness, hippocampal volume or CSF t-tau level; N+, presence of neurodegeneration or neuronal injury determined by cortical thickness, hippocampal volume or CSF t-tau level;”  into  “N−, absence of neurodegeneration or neuronal injury determined by cortical thickness or hippocampal volume; N+, presence of neurodegeneration or neuronal injury determined by cortical thickness or hippocampal volume;”.

I modified the frequency (%) of ATN in Tables 5 and 6 based on the new analysis.

Point 5: showing group level differences (are error bars SD or SEM?) misses the point of validation. The authors must report how each subject is classified according to KBASE values, or use internal cross validation to have some objective assessment.

Response 5: I agree with you. The inclusion and exclusion criteria for ADD, MCI, and CU in the KBASE-V were the same as those in the KBASE. I added the statistical comparisons between each clinical group and its corresponding group in the main KBASE cohort. I added Table 2 and modified the manuscript as follows.

Based on your comments, I added “The inclusion and exclusion criteria for ADD, MCI, and CU in the KBASE-V were the same as those in the KBASE. In addition, the CU group was divided into the CN group and the SCD group using the SCD criteria established by the SCD-Initiative group [4] in the KBASE-V.” on lines 122-125 of page 3, and “This cutoff value was the same as the previous study from the KBASE [59].” on lines 218-219 on page 5. I changed “reference 59 Villeneuve, S.; Rabinovici, G.D.; Cohn-Sheehy, B.I.; Madison, C.; Ayakta, N.; Ghosh, P.M.; La Joie, R.; Arthur-Bentil, S.K.; Vogel, J.W.; Marks, S.M., et al. Existing Pittsburgh Compound-B positron emission tomography thresholds are too high: statistical and pathological evaluation. Brain 2015, 138, 2020-2033, doi:10.1093/brain/awv112” into “Hwang, J.Y.; Byun, M.S.; Choe, Y.M.; Lee, J.H.; Yi, D.; Choi, J.W.; Hwang, S.H.; Lee, Y.J,; Lee, D.Y.; KBASE Research Group. Moderating effect of APOE ε4 on the relationship between sleep-wake cycle and brain β-amyloid. Neurology 2018, 90, e1167-e1173, doi:10.1212/WNL.0000000000005193” on lines 715-717 of page 16.

I added Table 2 and “There was no significant difference in age, MMSE, SBT, CDR, Stroop test, clock drawing, and the prevalence of APOE ε4 carriers and A+ between each clinical group and its corresponding group in the KBASE (Table 2). The education level in the CU and ADD groups of KBASE-V was lower than the corresponding group of KBASE. In comparison to the corresponding group of KBASE, the prevalence of women was higher in the KBASE-V CU group and lower in the KBASE-V MCI group. The MCI group of KBASE-V had higher CERAD and lower CDR-SB scores than the MCI group of KBASE. The CU and ADD groups of KBASE-V showed higher GDS and SMCQ scores than the corresponding group in the KBASE.” on lines 286-293 of page 7.  

 Table 2. Baseline characteristics of the KBASE and KBASE-V participants

Variables

(range)

CU of KBASE

CU of

KBASE-V

MCI of KBASE

MCI of

KBASE-V

ADD of

KBASE

ADD of

KBASE-V

n

291

167

139

72

87

56

Age,   years

69.2   (8.1)

67.7   (7.8)

73.7   (7.0)

73.1   (8.4)

73.0   (8.1)

75.6   (7.5)

Female

149   (51%)

103   (62%)a

92   (66%)

33   (46%)a

60   (80%)

44   (79%)

Education,   years

11.8   (4.8)

10.6   (5.1)b

10.0   (4.5)

8.9   (4.2)

9.1   (5.4)

5.7   (4.1)b

APOE   ε4 carrier

53   (18%)

35   (21%)

47   (34%)

16   (22%)

52   (60%)

26   (46%)

MMSE   (0-30)

26.9   (2.6)

26.5   (3.0)

22.5   (3.2)

22.7   (4.3)

16.4   (4.2)

16.0   (3.7)

CDR 0:   0.5: 1§

291 :   0: 0

167:   0: 0

0:   139: 0

0: 72:   0

0: 29:   58

0: 19:   37

CDR-SB   (0-18)§

0.01   (0.06)

0.01   (0.07)

1.50   (0.65)

1.16   (0.86)

5.05   (1.45)

4.81   (2.03)

SBT   (0-28)§

2.5   (3.3)

3.1   (4.4)

9.1   (6.3)

8.6   (6.7)

18.9   (5.3)

19.3   (6.4)

CERAD,   Z score

0.35   (0.56)

0.33   (0.47)

-0.98   (0.63)

-0.73   (0.59)b

-1.60   (0.71)

-1.59   (0.69)

SMCQ   (0-14)§

2.1   (2.0)

3.9   (3.4)*

5.6   (3.1)

6.0   (3.6)

5.7   (3.0)

8.8   (3.8)

GDS   (0-30)§

4.7   (4.9)

8.4   (6.5)*

10.3   (7.0)

10.4   (6.9)

7.9   (5.5)

11.9   (7.9)

LM   immediate recall

19.1   (7.3)

17.0   (8.1)*

9.2   (5.7)

7.7   (5.3)

3.8   (5.0)

2.5   (2.9)

LM   delayed recall

13.9   (7.4)

12.0   (7.8)*

3.9   (4.6)

3.4   (4.0)

0.7   (4.3)

0.5   (1.3)

RCFT   copy (0-36)

29.6   (6.2)

31.1   (6.6)*

24.7   (8.4)

27.1   (9.5)

17.8   (11.5)

14.8   (11.0)

Clock   drawing (0-30)

27.7   (2.8)

27.6   (3.2)

25.6   (4.9)

24.8   (5.8)

20.9   (7.4)

17.2   (7.9)

Stroop   test, Z score

0.18   (0.98)

0.12   (1.00)

-0.66   (0.98)

-0.55   (1.05)

-1.26   (1.15)

-1.34   (1.07)

COWAT

25.6   (11.4)

23.6 (10.4)

19.3   (9.0)

16.6   (9.5)

16.3   (10.0)

8.0   (7.8)

A+

52   (18.1%) (n=288)

26   (19.8%)

(n=131)

71   (53.0%)

(n=134)

28   (48.3%)

(n=58)

66   (81.5%)

(n=81)

36   (90.0%)

(n-40)

Data are shown as the mean (SD) or number (%). CU, cognitively unimpaired; MCI, mild cognitive impairment; ADD, Alzheimer’s disease dementia; KBASE, Korean Brain Aging Study for the Early Diagnosis and Prediction of AD; KBASE-V, an independent validation cohort of KBASE; APOE, apolipoprotein; MMSE, Mini-Mental State Examination; CDR, Clinical Dementia Rating scale; CDR-SB, CDR-Sum of Boxes; SBT, Short Blessed Test; CERAD, Consortium to Establish a Registry for AD; SMCQ, Subjective Memory Complaints Questionnaire; GDS, Geriatric Depression Scale; LM, Logical memory; RCFT, Rey Complex Figure Test; COWAT, Controlled Oral Word Association Test; A+, presence of amyloid pathology determined by abnormal amyloid PET finding in the KBASE and by abnormal amyloid PET finding or CSF Aβ42 level below cut point in the KBASE-V. a, p < 0.05 vs. its corresponding group of KBASE by chi-square test; b, p < 0.05 vs. its corresponding group of KBASE by student t-test; *p < 0.05 vs. KBASE CU group from analysis of covariance (ANCOVA) with sex and education level as covariates; p < 0.05 vs. KBASE MCI group from ANCOVA with sex as a covariate; p < 0.05 vs. KBASE ADD group from ANCOVA with education level as a covariate; §Increases in scores represent worsening.

On lines 431-449 of page 13 in the discussion section, I changed “The KBASE-V is a suitable cohort to validate potential AD biomarkers and to identify risk and prognostic factors of AD. A total of 216 (73.2%) participants underwent amyloid PET. The presence or absence of amyloid pathology according to CSF analysis or amyloid PET was confirmed in 229 (77.6%) participants. The percentage of participants in KBASE-V for whom amyloid biomarkers were examined was not less than in the previous cohort [49,65]”  into  “Each clinical group of KBASE-V showed generally comparable baseline characteristics with its corresponding group of KBASE, with some differences. General cognitive function such as MMSE, SBT, and CDR in each clinical group was similar between the KBASE and KBAE-V. Age and the prevalence of APOE ε4 carriers were also similar between each clinical group of KBASE-V and its corresponding group of KBASE. The education level in the CU and ADD groups of KBASE-V was lower than the corresponding group of KBASE. The cause is that the KBASE-V is a nationwide cohort and the KBASE is a cohort of highly educated people living in the capital and surrounding areas. A higher GDS score in the CU and ADD groups of KBASE-V may also be related to the socioeconomic differences between the KBASE and KBAE-V. The CERAD score in the CU and ADD groups of KBASE-V was similar to that in the corresponding group of KBASE. The cognitive function of the MCI group of KBASE-V appears to be slightly better than the corresponding group of KBASE, as the MCI participants in KBASE-V showed higher CREAD and lower CDR-SB scores than the MCI group in KBASE. The prevalence of A+ was similar between each clinical group of KBASE-V and its corresponding group of KBASE. A total of 216 (73.2%) participants underwent amyloid PET in the KBASE-V. The presence or absence of amyloid pathology according to CSF analysis or amyloid PET was confirmed in 229 (77.6%) participants of the KBASE-V. The percentage of participants in KBASE-V for whom amyloid biomarkers were examined was not less than in the previous cohort [49,65]. The KBASE-V is a suitable cohort for reconfirming potential new AD biomarkers such as blood biomarkers identified in the KBASE and identifying the risk and prognostic factors of AD.”.

Point 6: there is no intermediate precision reported for the CSF measures.

Response 6: Base on your comments, I added “We determined the analytical precision using 9 pooled CSFs and found that the mean (SD) between-run precision (percent coefficient of variation [%CV] through 5 runs) for CSF Aβ42, t-tau and p-tau was 9.1 (2.0), 10.9 (2.5) and 10.7 (6.2)%. All runs followed our standard operating procedure including the pre-defined run acceptance criteria including enough bead count (>20) and low %CV (< 25%) in duplicates of calibrators, aqueous QC and pooled CSFs.” on lines 246-251 of page 5-6.

Round  2

Reviewer 1 Report

The revised manuscript is an improvement, but the authors seem to have misunderstood a couple of my points. They need to be addressed.

Re: Point M2.1.
The authors' chosen post hoc group-comparison analysis has changed from LSD to an unspecified, Bonferroni-corrected statistical test.

– Which test is this? Justify your choice.

Re: Point M2.2

The authors have not answered my question.

The sentence (line 290, section 2.8) implies that the controls are age-matched to the other groups (MCI, ADD), but the manuscript also says that the control group is statistically younger than the ADD group.

If the controls are indeed younger, then in what sense are they "age-, sex-, and education-matched"? Matched with whom? This needs to be clear in the manuscript.

Two comparisons are interesting here: between cohorts (KBASE vs KBASE-V: table 2), within diagnostic group (e.g. CN); and within the KBASE-V cohort (table 3), between diagnostic groups, i.e., CN vs ADD (etc.).

Re: Point m3.

The authors should add to the manuscript (Line 977, p14) that depression was a key covariate in this group comparison. The methods section should also clearly state that depression was included in the group comparisons of cognition.

Author Response

Re: Point M2.1.
The authors' chosen post hoc group-comparison analysis has changed from LSD to an unspecified, Bonferroni-corrected statistical test.

– Which test is this? Justify your choice.

Response M2.1: I have consulted with a doctor of statistics. She recommended Bonferroni post hoc analysis for correction of multiple comparisons instead of LSD because Bonferroni adjustment can correct more conservatively multiple comparisons. To set a more stringent P-value when making multiple comparisons, the most commonly used method for adjusting the significance level for multiple pairwise comparisons is the Bonferroni corrections. You simply divide the probability threshold that you would use if you were performing a single test (usually 0.05) by the number of pairwise comparisons you are performing. For example, if you are performing three pairwise comparisons, you would reject the null hypothesis only if p ≤ 0.017(0.05/3=0.017). If you were performing four pairwise comparisons, you would reject the null hypothesis only if p ≤ 0.013 (0.05/4 = 0.013). In our study, we rejected the null hypothesis only if p ≤ 0.0083 (0.05/6 = 0.008333…).  (Reference: Katz, M.H. Study design and statistical analysis. A practical guide for clinicians; Cambridge University Press: Cambridge, UK, 2006; pp. 90.)

Re: Point M2.2

The authors have not answered my question.

The sentence (line 290, section 2.8) implies that the controls are age-matched to the other groups (MCI, ADD), but the manuscript also says that the control group is statistically younger than the ADD group.

If the controls are indeed younger, then in what sense are they "age-, sex-, and education-matched"? Matched with whom? This needs to be clear in the manuscript.

Two comparisons are interesting here: between cohorts (KBASE vs KBASE-V: table 2), within diagnostic group (e.g. CN); and within the KBASE-V cohort (table 3), between diagnostic groups, i.e., CN vs ADD (etc.).

Response M2.2: The CERAD battery and Stroop test had been administered to hundreds of healthy Korean elderly to get the normative data including Z scores in previous studies and the normative data had been presented in previous reports [references 18 and 41]. We applied the normative data from the previous reports in this KBASE-V study.   

On lines 266-268 of page 6, I changed “The Z scores were based on the mean and SD of each measure in the age-, sex- and education-matched normal population” into “The Z scores were based on the mean and SD of each measure in the age-, sex- and education-matched normal Korean elderly population reported in previous studies [18,41]”.

Re: Point m3.

The authors should add to the manuscript (Line 477, p14) that depression was a key covariate in this group comparison. The methods section should also clearly state that depression was included in the group comparisons of cognition.

Response m3: Based on your comments, I included GDS score as an additional covariate in the analysis of clinical and neuropsychological continuous variables. According to the new analysis, I modified Table 3 and 4, and the manuscript as follows.

On lines 262-266 of page 6, I changed “For comparison among the groups, we used chi-square tests for categorical variables, and one-way analysis of variance (ANOVA) for continuous variables and Z scores of neuropsychological tests” into “For comparison among the groups, we used chi-square tests for categorical variables and one-way analysis of variance (ANOVA) for continuous variables. When a statistical significant overall difference was detected in the ANOVA test, pairwise comparisons on means between diagnosis groups were conducted by Tukey method. Analysis of covariance (ANCOVA) was performed for statistical analysis of Z scores of neuropsychological tests with GDS score as a covariate”.

On lines 269-273 of page 6, I changed “Analysis of covariance (ANCOVA) was performed for statistical analysis of BMI, ASMI, hand grip, gait speed, MNA, total sleep time, ESS, CCI, MMSE, SBT, SMCQ, MAC-Q, GDS, NPI, BDS-ADL, CDR-SB, and raw scores of neuropsychological tests with age, sex, and education level as covariates”  into  “GDS score was compared using ANCOVA with age, sex, and education level as covariates. ANCOVA was also performed for statistical analysis of BMI, ASMI, hand grip, gait speed, MNA, total sleep time, ESS, CCI, MMSE, SBT, SMCQ, MAC-Q, NPI, BDS-ADL, CDR-SB, and raw scores of neuropsychological tests with age, sex, education level, and GDS as covariates”.

On lines 321-322 of page 8, I changed “NPI scores and the prevalence of APOE ε4 carriers and sarcopenia were significantly higher in the ADD group than in the other groups. The BDS-ADL scores were significantly higher in the MCI group than in the CN group”  into  “The NPI and BDS-ADL scores and the prevalence of APOE ε4 carriers and sarcopenia were significantly higher in the ADD group than in the other groups”.

On lines 350-361 of page 10, I changed “The Z scores of naming and immediate and delayed recalls of word list, immediate recall of LM, constructional praxis recall, RCFT delayed recalls, and correct color reading of Stroop test were significantly different among the four groups….. The scores of verbal fluency and digit span backward test in the SCD group were no different from those in the MCI group, but were significantly lower than in the CN group. The scores of word list recognition, clock drawing and FAB were significantly different between all groups, except between the CN and SCD groups. …….The scores of constructional praxis, COWAT and TMT-B in the CN group were significantly higher in those of the ADD and MCI groups, but were not different from those in the SCD group”  into  “The scores of naming, immediate recall of LM, constructional praxis recall, and 30 min delayed recall of RCFT were significantly different among the four groups…. The scores of verbal fluency, digit span backward, correct color reading of Stroop test, and TMT-B in the SCD group were no different from those in the MCI group, but were significantly lower than in the CN group. The scores of word list immediate and delayed recalls, word list recognition, 3 min delayed recall of RCFT, and FAB were significantly different between all groups, except between the CN and SCD groups. …. The scores of constructional praxis, clock drawing, COWAT, and LM recognition in the CN group were significantly higher in those of the ADD and MCI groups, but were not different from those in the SCD group.”

On lines 484-486 of page 14, I added “We included depression score as a covariate in the analysis to compare the clinical and neuropsychological variables among the groups because depression was a key covariate in this group comparison”.

On lines 490-491 of page 14, I changed “The CDR-SB scores and ADL impairment began to increase significantly from the MCI stage”  into  “The CDR-SB scores began to increase significantly from the MCI stage and ADL impairment began at the mild ADD stage”.